



# Gravity wave induced cross-isentropic mixing: A DEEPWAVE case study

Hans-Christoph Lachnitt[1], Peter Hoor[1], Daniel Kunkel[1], Martina Bramberger[2], Andreas Dörnbrack[3], Stefan Müller[1,4], Philipp Reutter[1], Andreas Giez[3], Thorsten Kaluza[1], and Markus Rapp[3]

[1]Institute for Atmospheric Physics, Johannes Gutenberg University Mainz, Germany
[2]NorthWest Research Associates, Boulder, USA
[3]Institute of Atmospheric Physics, German Aerospace Center (DLR) Oberpfaffenhofen, Germany
[4]now at enviscope GmbH, Frankfurt/Main, Germany

**Correspondence:** Lachnitt, Hans-Christoph (hlachnit@uni-mainz.de)

**Abstract.** Orographic gravity waves (i.e. mountain waves) can potentially lead to cross-isentropic fluxes of trace gases via the generation of turbulence. During the DEEPWAVE (Deep Propagating Gravity Wave Experiment) campaign in July 2014 we performed tracer measurements of carbon monoxide (CO) and nitrous oxide ($N_2O$) above the Southern Alps during periods of gravity wave activity. The measurements were taken along two stacked levels at 7.9 km in the troposphere and 10.9 km in the stratosphere. A detailed analysis of the observed wind components shows, that both flight legs were affected by vertically propagating gravity waves with momentum deposition and energy dissipation between the two legs. Corresponding tracer measurements indicate turbulent mixing in the region of gravity wave occurrence.

For the stratospheric data we identified mixing leading to a change of the cross-isentropic tracer gradient from the upstream to the downstream region of the Southern Alps. Based on the quasi-inert tracer $N_2O$ we identified two distinct layers in the stratosphere with different chemical composition on different isentropes. The CO-$N_2O$ relationship clearly indicates that irreversible mixing between these two layers occurred. Further we found a significant change of the $N_2O$-$\Theta$-profiles from the upstream to the downstream side above the Southern Alps with different gradients of the $N_2O$-$\Theta$-relation just above the tropopause. A scale-dependent gradient analysis reveals that this gradient change is triggered in the region of gravity wave occurrence.

The power spectra of the in-situ measured vertical wind, $\Theta$, and $N_2O$ indicate the occurrence of turbulence above the mountains associated with the gravity waves in the stratosphere. The estimated eddy dissipation rate based on the measured three dimensional wind indicates a weak intensity of turbulence in the stratosphere above the mountain ridge. The $N_2O$-$\Theta$-relation downwind the Alps modified by the gravity wave activity provides clear evidence that trace gas fluxes, which were deduced from wavelet co-spectra of vertical wind and $N_2O$ are at least in part cross-isentropic.

Our findings thus indicate that orographic waves led to turbulent mixing on both flight legs in the troposphere and in the stratosphere. Despite only weak turbulence during the stratospheric leg, the cross isentropic gradient and the related composition change on isentropic surfaces from upstream to downstream the mountain unambiguously conserves the effect of turbulent mixing by gravity wave activity before the measurements. This finally leads to irreversible diabatic trace gas fluxes and thus has a persistent effect on the UTLS trace gas composition.



# 1 Introduction

Orographic gravity waves play an important role for the thermal and dynamical structure of the atmosphere and may affect the large scale stratospheric circulation (e.g. Smith et al., 2008; Fritts and Alexander, 2003; Kim et al., 2003; Rapp et al., 2021).

Many observations of stratospheric gravity waves are based on satellite observations (e.g. Alexander et al., 2008; Geller et al., 2013; Ern et al., 2018). However, in the upper troposphere/lower stratosphere (UTLS) observations of gravity waves from aircraft and balloon soundings are essential for process studies beyond the resolution of satellites (e.g. Podglajen et al., 2016; Krisch et al., 2017; Wright and Banyard, 2020; Rapp et al., 2021). They propagate across the UTLS where static stability increases at the tropopause (Woods and Smith, 2010). Gravity waves may locally lead to the generation of convective instabilities

(Lane and Sharman, 2006) or dynamical instability induced by vertical shear of the horizontal wind (Panofsky et al., 1968; Kunkel et al., 2014; Söder et al., 2021). Both may lead to the occurrence of turbulence, particularly when wave breaking occurs with potential subsequent mixing of trace species (Pavelin et al., 2002; Whiteway et al., 2003).

The tropopause as a central feature of the UTLS acts as a dynamical barrier for transport of species and the formation of trace gas gradients at the tropopause (e.g. Gettelman et al., 2011; Woiwode et al., 2018; Hoor et al., 2002; Pan et al., 2004). To

overcome this dynamical barrier diabatic processes are required, which can be associated with radiation driven processes or phase transitions of water. In addition turbulence occurrence, associated with wind shear above the tropopause (Shapiro, 1980; Söder et al., 2021; Kaluza et al., 2021; Lilly et al., 1974) provides an efficient process for cross-isentropic mixing and thus irreversible trace gas exchange at the tropopause. Turbulence at the tropopause can be generated by a number of processes, e.g. strong shear at the jet streams (e.g. Kaluza et al., 2021), convection (Wang, 2003; Mullendore et al., 2005; Homeyer et al.,

2017) or frontal uplift leading to strong vertical convergence of the horizontal wind (Panofsky et al., 1968; Söder et al., 2021) and the generation of gravity waves (e.g., Whiteway et al., 2003; Wang, 2003).

The effect of orographic waves on tracer transport and mixing and the emerging tracer redistributions has been frequently discussed (Schilling et al., 1999; Heller et al., 2017; Moustaoui et al., 2010; Zhang et al., 2015; Podglajen et al., 2017; Grubišić et al., 2008; Woiwode et al., 2018). Observational studies mainly addressed the analysis of local kinematic fluxes using the co-

variance of the local vertical winds and the tracer fields (Shapiro, 1980; Schilling et al., 1999). Based on airborne observations in a tropopause fold Shapiro (1980) identified ozone and particle fluxes in regions of turbulence occurrence and shear. They estimated vertical turbulent tracer fluxes by approximating the change of the mean tracer flux by the divergence of the turbulent vertical tracer flux. This approach has been used to derive mountain wave induced tracer fluxes using the flux divergence approach between two flight legs (with altitude differences of typically 1000 m) providing evidence for a vertical flux of carbon

monoxide (Schilling et al., 1999) or water vapour (Heller et al., 2017).

Direct observations of gravity wave induced vertical cross-isentropic tracer transport are sparse, since this requires high resolution measurements of passive tracers (i.e. tracers without chemical or microphysical sources or sinks) exactly in the region of turbulence occurrence associated with these waves. Kinematic fluxes based on the covariance of the local vertical wind and





the passive tracers are not necessarily irreversible. Analysis of the relation between tracer mixing and orographic wave activity (Moustaoui et al., 2010; Mahalov et al., 2011) highlighted the importance of the local tropopause structure for the interpretation of fluxes. Moustaoui et al. (2010) concluded on the basis of in-situ observations that the layers within the UTLS including the tropopause may behave like material surfaces and that vertical displacements are largely reversible despite the occurrence of

gravity waves. They further pointed out that the observed tracer characteristics are the result of a non-linear interaction between synoptic-scale waves modulated by shorter gravity waves, leading first to reversible transport and tracer mixing.

This study steps in here to provide evidence on the basis of observed passive tracers that the breaking of mountain waves can lead to diabatic tracer redistribution in the tropopause region by cross-isentropic mixing. We will investigate how orographic gravity wave induced turbulence leads to a non-local persistent effect on the UTLS composition downwind the turbulent mixing

region. The paper is organised as follows: First we will describe the measurements and techniques, and will briefly introduce the DEEPWAVE project and the meteorological conditions during the flight. Section 3 will focus on the observations and the identification of mixing from tracer measurements of $N_2O$ and CO. In section 4 we will identify the cross-isentropic mixing and use different methods to show that gravity wave induced mixing has affected the observed tracer-$\Theta$ relationship with a persistent impact on the composition in the downstream side of the mountain ridge. The results will be analyzed for non-isentropic

transport followed by a discussion.

## 2   Data and Methods

### 2.1   Campaign and instrumentation

The measurements were performed in the frame of the DEEPWAVE (Deep Propagating Gravity Wave Experiment) mission in summer 2014 over New Zealand (Fritts et al., 2016; Gisinger et al., 2017). The project combined a comprehensive set of

ground-based measurements as well as airborne measurements by the NSF (National Science Foundation)/NCAR (National Center for Atmospheric Research) Gulfstream GV HIAPER (High-performance Instrumented Airborne Platform for Environmental Research) and German DLR (Deutsches Zentrum für Luft- und Raumfahrt) Dassault Falcon 20E aircraft. Airborne measurements were carried out from Christchurch during June and July 2014 and covered the upper troposphere and the lower stratosphere providing remote sensing data up to 80 km. The majority of the measurements focused on the Southern Alps to

study the evolution of mountain waves from their source to the breaking regions in the mesosphere. The two aircraft partly performed coordinated flights in the tropopause region to study the propagation and potential dissipation of gravity waves in this region.

### 2.2   Aircraft Measurements

Tracer measurements of $N_2O$ and CO were performed using the 'University MAinz Quantum Cascade Absorption Spectro-

meter (UMAQS, Müller et al., 2015). The instrument consists of a Harriott cell with a path length of 36 m. We detected $N_2O$ and CO at wavenumbers of 2002.75 cm$^{-1}$ and 2003.16 cm$^{-1}$, respectively, by scanning with a frequency of 9 kHz across





the absorption lines at a cell pressure of 50 hPa. The instrument is capable of simultaneously measuring $N_2O$ and CO with a temporal resolution of 10 Hz ultimately limited by the flow speed and purging time of the cell. The instrument is in-situ calibrated using secondary standards of compressed ambient air which are traced to primary standards of NOAA prior and after the campaign. The precision of the data (given here for the 1 Hz averaged data) is on the order of 0.05 ppbv (1-$\sigma$) for

$N_2O$ and CO, respectively.

We further use measurements of the basic atmospheric state parameters at 10 Hz provided by the Falcon aircraft (Krautstrunk and Giez, 2012). The three-dimensional wind is deduced from a 5-hole gust probe located in the noseboom providing true air speed and the ground speed from the inertial system (Mallaun et al., 2015). For the vertical wind component an uncertainty of ±0.3 m s$^{-1}$ is estimated. The true static air temperature is provided with a measurement uncertainty of ±0.5 K.

**2.3  Meteorological analysis data**

To support our analysis and obtain information of the synoptic background we used ECMWF IFS (Integrated Forecasting System) operational analysis data. The data are available every six hours and have been interpolated on a horizontal grid with a spacing of 0.125° for our analysis. In total the model has 137 vertical hybrid sigma-pressure levels from the surface up to 0.01 hPa. The vertical grid spacing is about 300 m in the tropopause region. For our analysis the model data are linearly

interpolated in time and space onto the location of the flight track.

**2.4  Wavelet analyses**

The wavelet analysis is a tool for the spectral analysis of time series data (Torrence and Compo, 1998). The advantage over Fourier analysis is the ability to show information in time and frequency space. Wavelet analysis has frequently been used for the analysis of gravity waves (e.g., Bramberger et al., 2017; Zhang et al., 2015; Woods and Smith, 2011, 2010). The

wavelet basis function which we use in this study is the Morlet wavelet with non-dimensional frequency $\omega_0$=6 as defined in Torrence and Compo (1998). To reveal periods with significant wavelet power we determined the 95% confidence level in the respective analyses below as described in Torrence and Compo (1998). The wavelet cospectrum can be used to analyse spectral characteristics of turbulent fluxes (e.g. Mauder et al., 2007; Zhang et al., 2015). The wavelet cospectrum $W^{AB}$ of two time series $A$ and $B$ with the wavelet transforms $W^A$ and $W^B$ is defined as

$$W_n^{AB}(s) = \Re\left\{W_n^A(s)W_n^{B*}(s)\right\} \tag{1}$$

where the asterisk denotes the complex conjugation, $n$ a local time index and $s$ the wavelet scale (Torrence and Compo, 1998; Liu et al., 2007).

Another tool which we use in this study is the wavelet coherence $R^2$. It represents the covariance between the two time series $A$ and $B$ and can be used to identify frequency and time intervals with strong and weak coherence between the variations of

the two data sets. The wavelet coherence is defined as

$$R_n^2(s) = \frac{\left|S\left(s^{-1}W_n^{AB}(s)\right)\right|^2}{S\left(s^{-1}\left|W_n^A(s)\right|^2\right) \cdot S\left(s^{-1}\left|W_n^B(s)\right|^2\right)} \tag{2}$$





where $S$ is a smoothing operator in space and time (for further details, see e.g., Torrence and Compo, 1998; Grinsted et al., 2004; Jevrejeva et al., 2003).

## 3 Observations and meteorological situation during FF09

### 3.1 Meteorological situation on 12. July 2014

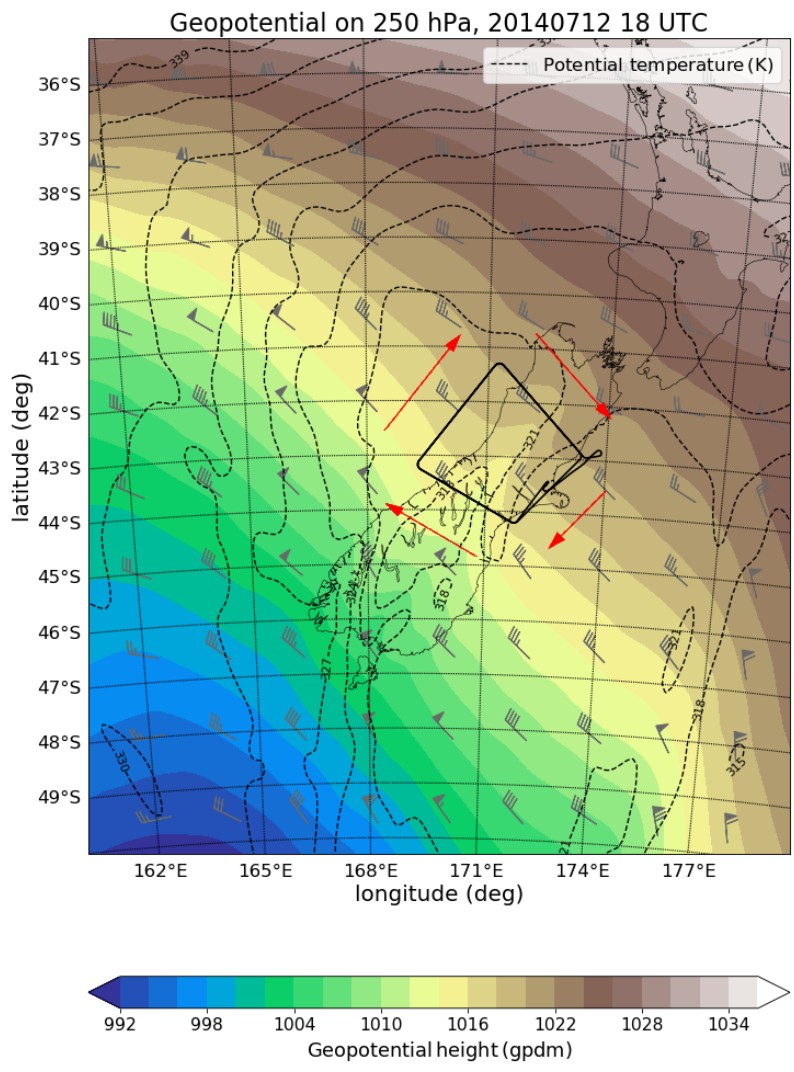

**Figure 1.** Geopotential height (color) and potential temperature (dashed) at the 250 hPa surface with the flight track for research flight FF09 during 12. July 2014, 18:00 UTC. The red arrows indicate the flight direction.





In this study we focus on research flight FF09 of the Falcon aircraft on the 12. July 2014 starting at 17:15 UTC to 20:15 UTC (Fig. 1). The goal of this flight was to investigate the dynamical and chemical structure of the atmosphere in the vicinity of tropopause during a mountain wave event (Gisinger et al., 2017; Smith et al., 2016). As can be seen in Fig. 1 a rectangular pattern was flown to measure different wave responses in the northern and middle part of the Southern Island of New Zealand

at two different pressure levels (330 hPa and 260 hPa, Fig. 2) corresponding to approximately 7.9 km and 10.9 km pressure altitude. The time between two vertically stacked legs at the same location was 75 minutes (Fig. 3).

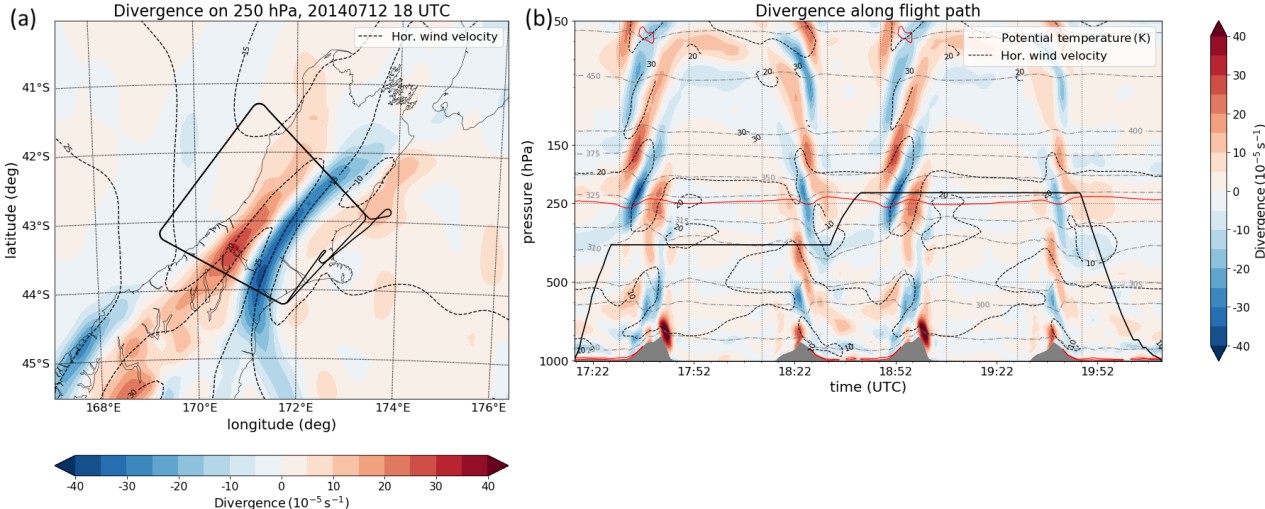

**Figure 2.** Divergence of the horizontal wind during the time of flight a) at 250 hPa and b) as vertical cross section along the flight track indicating the signature of gravity waves over the Southern Alps. The solid red line denotes the -2 pvu isoline, the black dashed lines denotes contours of the horizontal wind velocity (10, 15, 20, 25 m s$^{-1}$ in a) and 10, 20, 30 m s$^{-1}$ in b)) and the gray dashed lines in b) denotes contours of potential temperature.

According to Gisinger et al. (2017) the synoptic situation can be characterized by a trough located west of New Zealand with a weak surface low south of the Islands causing northwesterly winds in the troposphere (TNW regime, their Fig. 2e). In the upper level at 250 hPa at the eastern side of the approaching upper level trough a weak gradient of the geopotential height

led to a northwesterly flow with moderate horizontal winds of typically 20 m s$^{-1}$ along the southern flight legs (with 30 m s$^{-1}$ above the mountain ridge). The tropopause became relatively flat in the region and at the time of the measurements (Fig. 2b). These conditions led to the excitation of mountain waves and generated varying and moderate gravity wave responses over South Island (Gisinger et al., 2017). Figure 2a shows the divergence of the horizontal wind at 250 hPa at 18:00 UTC which corresponds roughly to the time of flight. The vertical cross section interpolated in time and space along the flight path shows a

wave pattern above the mountain ridge indicating the excitation and propagation of orographic gravity waves, which propagate deep into the stratosphere (Fig. 2b).

It is evident from Figure 2b that the upper flight leg at about 10.9 km is just above the dynamical tropopause. A close inspection of Fig. 2b reveals an almost constant altitude of the dynamical tropopause along the flight path without strong horizontal





gradients or folds in the region of our measurements.

As mentioned above, a rectangular flight pattern at two different altitudes (7.9 km and 10.9 km) was chosen for the measurement of gravity waves. Here we focus on the flight legs parallel to the wind and orthogonal to the mountains of research flight FF09.

## 3.2 Time series analysis of FF09

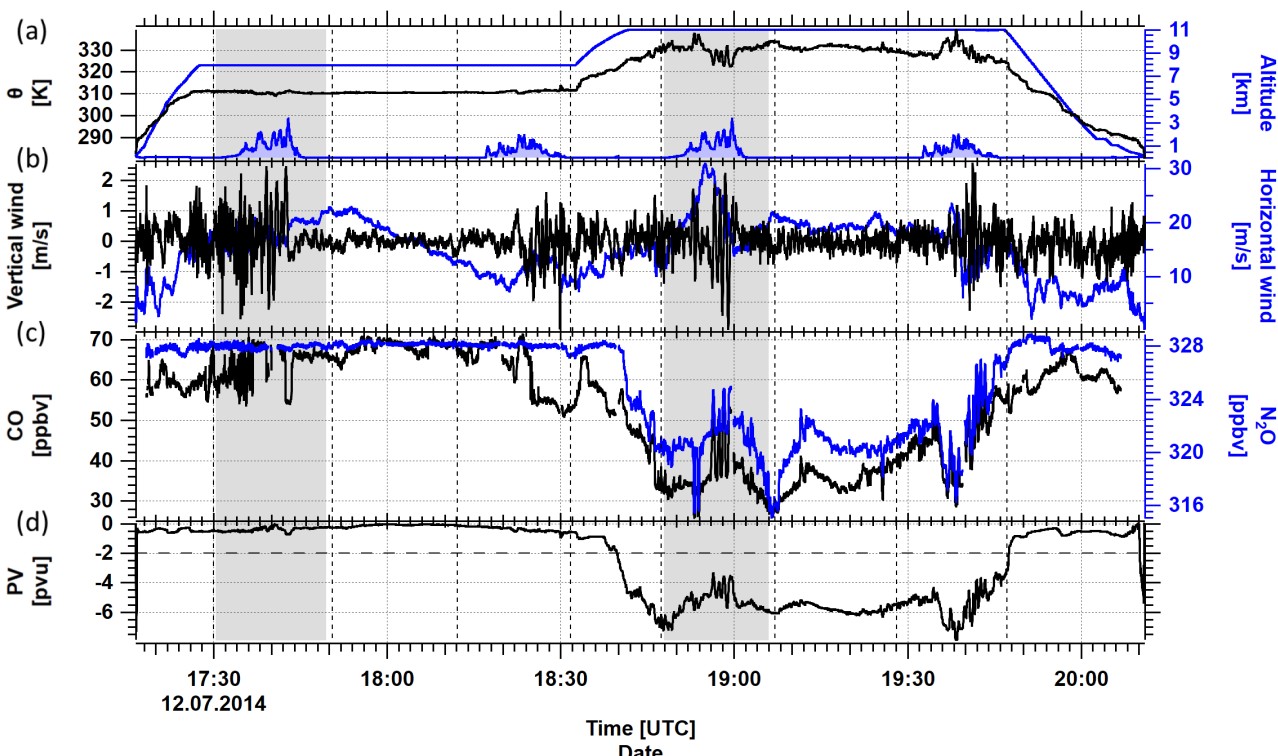

**Figure 3.** Time series of a) potential temperature $\Theta$ from the measurements (black), altitude (blue) above surface elevation (filled blue), b) vertical wind (black), horizontal wind (blue), c) $N_2O$ (blue) and CO (black) and d) potential vorticity interpolated along the flight track. The light gray box indicates the stratospheric flight section for the detailed mixing analysis. Vertical dashed lines mark turning points of the aircraft. Surface elevation was interpolated from SRTM15+ data (Tozer et al., 2019).

During flight FF09 on 12. July 2014 the flight legs were almost parallel to the horizontal wind and directed almost perpendicular to the Southern Alps. Pronounced fluctuations of the vertical wind occurred above the Southern Alps at each flight leg crossing the mountain ridge (Fig. 3).

10   The tropopause was crossed around 18:40 UTC, as indicated by the sharp decrease of the $N_2O$ mixing ratio and the analysed PV. Particularly, in the regions of strong variability of the vertical wind also $\Theta$, $N_2O$ and CO show enhanced variability and





strong fluctuations during the stratospheric part of the flight.

The fluctuations of potential temperature $\Theta$ reached an amplitude of $\Delta\Theta = 9$ K (Fig. 3). Corresponding oscillations of the vertical wind velocity reached 5 m/s peak to peak with associated variability of $N_2O$ on the order of 4 ppbv mirroring the os-cillations of potential temperature during both flight sections parallel to the wind. These features occurred above the mountains where the meteorological analysis shows a slight altitude variability of the dynamical tropopause (2 pvu, Fig. 2).

### 3.3 Orographic waves during FF09



**Figure 4.** Cross section of the two southern stacked flight legs crossing the Southern Alps showing $N_2O$ (green), $\Theta$ (blue) and vertical wind $w$ (black) for the upper leg at 10.9 km (top three panels) and the lower leg at 7.9 km with surface elevation (bottom). Both legs are separated by 75 minutes in time. The upper leg lies in the lower stratosphere, the lower leg lies in the upper troposphere just below the tropopause.





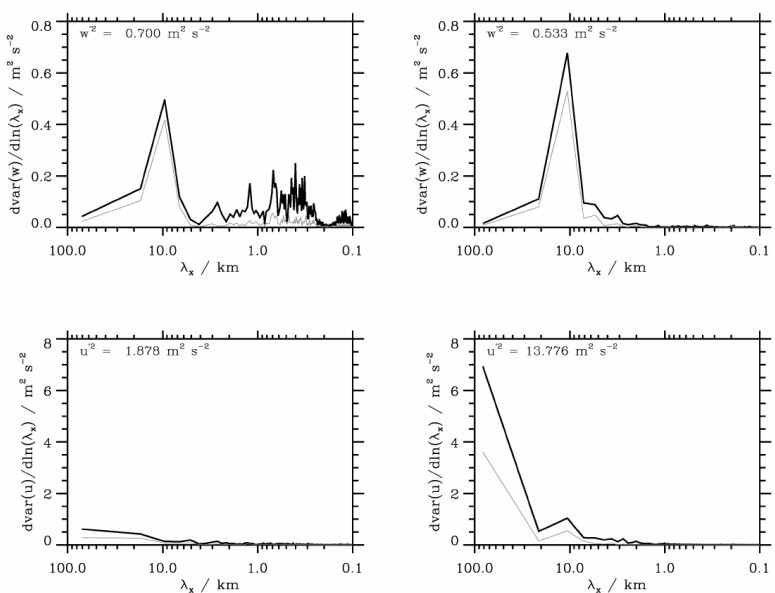

**Figure 5.** Binned energy spectra for the vertical horizontal wind component $w$ (top row) and the horizontal wind $V_H$ (bottom row) from the two southern flight legs of FF09, which crossed the mountains based on the 10 Hz data (corresponding to flight segments from 17:30-17:50, and 18:47-19:07 in Fig. 3). The left column shows the lower tropospheric flight track. Thick black lines are results without tapering window, thin black lines are spectra tapered with a Hanning window.

The flight sections of the two southern legs where strongly affected by orographic waves (Fig. 4). Both legs show strong fluctuations of the vertical wind component $w$ and the potential temperature $\Theta$ with amplitudes of $\pm 2$ ms$^{-1}$. The passive tracer nitrous oxide $N_2O$, which has a lifetime of 110 years in the lower stratosphere, indicates corresponding fluctuations at the upper level in the stratosphere. At the lower level $N_2O$ shows a weak variability. Due to its virtually constant abundance in

the troposphere $N_2O$ does not show corresponding oscillations to $w$ and $\Theta$ between 170.1°E-170.7°E. However, its variability does increase downwind the mountains similar to $w$ and $\Theta$, indicating the occurrence of turbulence. At the upper level such a breakdown of scales is not prominent, although the fluctuations of $\Theta$, $w$ and $N_2O$ (Fig. 4) are indicative for at least a kinematic flux of $N_2O$, but only weakly pronounced small scale variability of $w'$.

Before studying tracer transport and mixing we analyzed the dynamical properties of the orographic gravity waves. The binned

energy spectra of the vertical wind and the horizontal wind speed $V_H$ are shown in Fig. 5 for the southern flight legs crossing the mountains. Both legs show pronounced peaks in the $w$-spectra at about 10 km horizontal wavelength. For the lower leg higher values for the vertical turbulent kinetic energy occurred (as given by the squared variance of the vertical wind $\overline{w'^2} = 0.70$ m$^2$ s$^{-2}$, the overline denotes the average over the whole 200 km long flight leg) compared to the upper leg ($\overline{w'^2} = 0.53$ m$^2$ s$^{-2}$). However, this energy seems to reside in scales smaller than about 1 km. Therefore, the spectral amplitude

associated with the mountain waves at horizontal wavelengths of $\lambda_x = 10$ km is smaller at the lower leg, whereas at the upper leg wave motions with $\lambda_x \approx 10$ km dominate. No vertical energy is found at larger scales in both legs.





| | tropospheric leg | | | stratospheric leg | | |
|---|---|---|---|---|---|---|
| | upstream | mountain | downstream | upstream | mountain | downstream |
| $\overline{u'w'}$ / m$^2$s$^{-2}$ | 0.02 | -0.37 | -0.18 | -0.01 | -0.16 | 0.02 |
| $\overline{v'w'}$ / m$^2$s$^{-2}$ | 0.00 | -0.21 | 0.09 | 0.01 | 0.02 | -0.02 |
| $\overline{u'p'}$ / Wm$^{-2}$ | -0.24 | -12.48 | -8.72 | -0.89 | -24.80 | -1.52 |
| $\overline{v'p'}$ / Wm$^{-2}$ | 0.32 | 7.08 | 4.91 | 0.24 | 3.28 | 0.84 |
| $\overline{w'p'}$ / Wm$^{-2}$ | 0.04 | 4.17 | 0.07 | 0.04 | 1.08 | 0.02 |

**Table 1.** Wave momentum and wave energy fluxes for the two southern legs separated in upstream, downstream and across mountain legs according to Fig. 7.

The situation is different for the horizontal wind spectra where the energy is at longer horizontal scales. Only the upper leg shows a spectral peak at 10 km similar to the one in $w$. Thus, there are two distinct gravity wave modes: one with a long horizontal wavelength (also partly seen in $V_H$ in Fig. 3 around 17:45) that is a response to the airflow over the whole mountain range and one with $\lambda_x \approx 10$ km. The long mode is totally absent in the lower leg and corresponds well to the rather uniform horizontal wind as shown in Fig. 3 (17:30-17:50 UTC). Probably this long mode was not fully captured by the limited lengths of the legs as flown by the DLR Falcon. The other, shorter mode in the horizontal wind spectra is well-developed only in the upper leg. However, the increase of the spectral variance from the lower leg to the upper leg by a factor of seven from 1.88 m$^2$ s$^{-2}$ to 13.78 m$^2$ s$^{-2}$ must be mainly related to the long wave observed there according to Fig. 5. The downstream regions of the lower leg show increased variances of the vertical wind $w'$ at short horizontal wavelengths. This might be related to the increase of small-scale energy as can be seen in the spectra shown in Fig. 5. The specific zonal momentum fluxes $\overline{u'w'}$ are negative above the mountains and their magnitude is much larger than up- and downstream. This indicates a vertical upward transport of negative horizontal momentum that is characteristic of vertically propagating mountain waves.

An estimate of the vertical momentum flux divergence

$$-\frac{1}{\rho}\frac{\partial}{\partial z}\overline{\rho u'w'} \approx -\frac{\partial}{\partial z}\overline{u'w'} = \frac{\partial u}{\partial t} \tag{3}$$

yields a deceleration of the zonal flow $\partial u/\partial t$ of about 6 m s$^{-1}$ d$^{-1}$. This indicates momentum deposition by dissipating mountain waves most likely occurring in the layer between the lower leg and the upper leg. The slowdown does not seem to affect the wave-induced increase of horizontal wind in the upper flight leg. Thus, the momentum dissipation must have occurred between the two flight segments separated vertically by 3000 m. This argument is supported by the small-scale signatures found in all wind components downstream of the coherent waves in the lower leg. They indicate turbulent modes associated with local instabilities above this level.

The meridional momentum fluxes $\overline{v'w'}$ are much smaller than $\overline{u'w'}$ (Tab. 1) and will not be considered here. The zonal wave energy fluxes $\overline{u'p'}$ are negative in all segments of both legs (Tab. 1) and their magnitudes are largest directly over the mountains. Together with the positive vertical wave energy flux $\overline{w'p'}$ this finding suggests vertically propagating mountain waves that travel against the mean flow, therefore, $\overline{u'p'} < 0$.





It is interesting to note that the vertical wave energy flux $\overline{w'p'}$ decreases with height by a factor of four, which means that the waves are attenuated as they propagate from the lower leg to the upper leg. This supports the idea that dissipation must have occurred in the layer between 7.9 km and 10.9 km altitude. Vertical and horizontal wave energy fluxes are drastically reduced in the up- and downstream segments indicating no significant vertical wave propagation there.

5 **3.4 Observation of mixing**

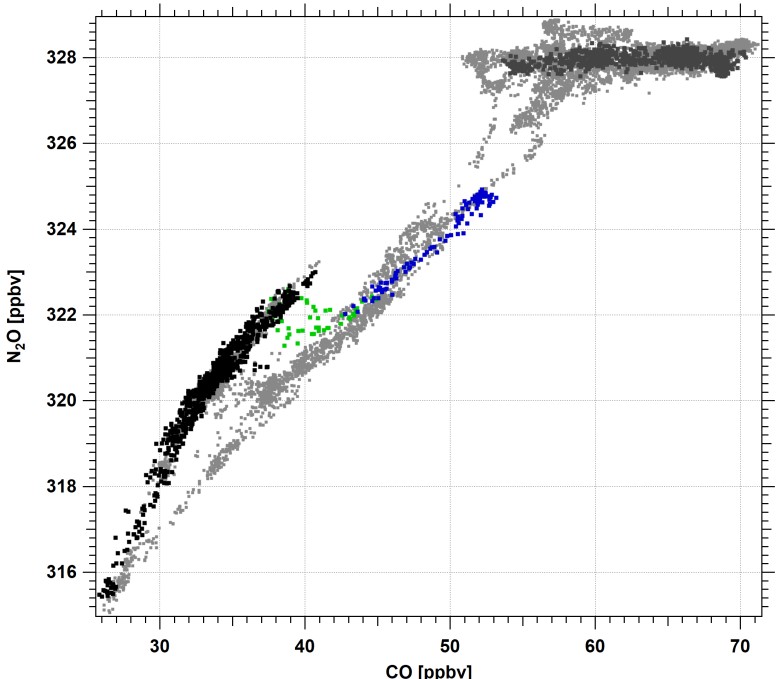

**Figure 6.** Scatterplot of $N_2O$ versus CO for FF09 on 12 July 2014. The light gray points show the correlation of $N_2O$ and CO for the whole flight. Colored data points denote the upper south-western flight leg from 18:48 UTC to 19:06 UTC in Figure 3 (also compare Fig. 7). Black colors indicate potential temperatures $\Theta > 328.1$ K, blue for $\Theta < 326.3$ K. The region between these two levels is marked in green. The lower leg lies entirely in the troposphere as indicated by the the dark gray data points of $N_2O=328$ ppbv.

We use tracer-tracer scatter plots of CO and $N_2O$ to investigate if mixing occurred in the region of the enhanced wave activity. Since $N_2O$ has no chemical sink in the atmosphere below 25 km and a lifetime of 110 years in the lower stratosphere, it is virtually homogeneously distributed in the troposphere, but exhibits a weak vertical gradient in the stratosphere (Müller et al., 2015). In contrast, CO has a chemical lifetime on the order of weeks to months in the tropopause region. Thus, it shows a
10 sharper gradient at the tropopause. Particularly, in the absence of mixing from the troposphere CO would fall to mixing ratios of 10-15 ppbv given by the balance of CO production from methane oxidation and faster CO degradation by OH. Any higher value is inevitably linked to a contribution of tropospheric air. Therefore mixing, in the sense of irreversible tracer transfer, can be detected by comparing CO to any long-lived tracer with a stratospheric gradient. The approach has been extensively used to





detect tropospheric influence in the stratosphere by using CO and ozone ($O_3$) (e.g. Hoor et al., 2002; Zahn and Brenninkmeijer, 2003; Pan et al., 2004). Here we use $N_2O$ instead of $O_3$, since it is purely controlled by atmospheric dynamics in the lower stratosphere due to the absence of local photochemical sources and sinks.

The scatter plot of CO versus $N_2O$ for both southwesterly legs of FF09 is shown in Fig. 6. The orographic waves at the lower leg appear at almost constant $N_2O$-levels of 328 ppbv. This is due to the fact, that in the troposphere no gradients of $N_2O$ are present. Thus mixing does not change the $N_2O$ mixing ratio as long as no stratospheric air is involved. Thus we cannot diagnose mixing for the lower leg on the basis of tracer-tracer correlations.

However, for the stratospheric legs across the mountain ridge during FF09 the scatter plot clearly shows different chemical regimes in the stratosphere (i.e. for $N_2O < 326$ ppbv, Fig. 6) as indicated by the two different branches of the two tracers. These two branches indicate two distinct air masses within the lower stratosphere which differ in their chemical composition as evident by the different $N_2O$ mixing ratios. A detailed analysis shows that the two branches of the correlation can be assigned to two distinct potential temperature intervals which correspond to two layers of different chemical composition for $\Theta > 328.1$ K and $\Theta < 326.3$ K. Notably, the data points which fall inbetween the two compact relations (and thus isentropes as given above) connect both air masses. They thus mark a layer between $\Theta < 328.1$ K and $\Theta > 326.3$ K, where the tracer-tracer diagram indicates mixing between the two branches (green crosses in Fig. 6).

To put the stratospheric part (i.e. for $N_2O < 327$ ppbv) of the tracer-tracer data of the scatter plot in a geophysical and meteorological context Fig. 7 shows the time series of potential temperature and $N_2O$ color coded according to the regimes identified from the scatter-plot (Fig. 6). The two branches of the correlation can be clearly assigned to different isentropes separating air masses with different chemical composition. Notably those points which indicate mixing in the tracer-tracer scatter plot fall inbetween the distinct layers. As evident from Fig. 7 the region where the chemical composition indicates mixing (green) corresponds to the occurrence of waves as indicated by strong fluctuations of the vertical wind and potential temperature. Therefore we hypothesize that mountain wave induced mixing must have occurred in the stratosphere leading to the observed tracer variability as shown above. In the following we will therefore focus on the stratospheric flight section across the mountains from 18:48 UTC to 19:06 UTC as indicated by the gray box in Fig. 3 if not noted differently.

## 4 Analysis of cross-isentropic mixing

Kinematic fluxes on the basis of the covariance of vertical wind and tracer variability $w'\chi'$ might provide information on the local vertical fluxes. For a correct estimate of an irreversible flux one needs to calculate the flux divergence (Shapiro, 1980). However, this would require simultaneous measurements of the tracer of interest on two vertically closely stacked levels, which can not be accomplished with one aircraft. A comparison of the local fluxes for the stacked levels is in principle possible. However, due to the large vertical spacing of 3 km potential influence from large scale horizontal advection could strongly impact the flux divergence estimates between the two levels.

Adiabatic vertical displacements of air masses may simultaneously displace the location of isentropes and tracer isopleths, which therefore does not lead to irreversible tracer transport and mixing at a given location downwind of a mountain ridge (i.e.

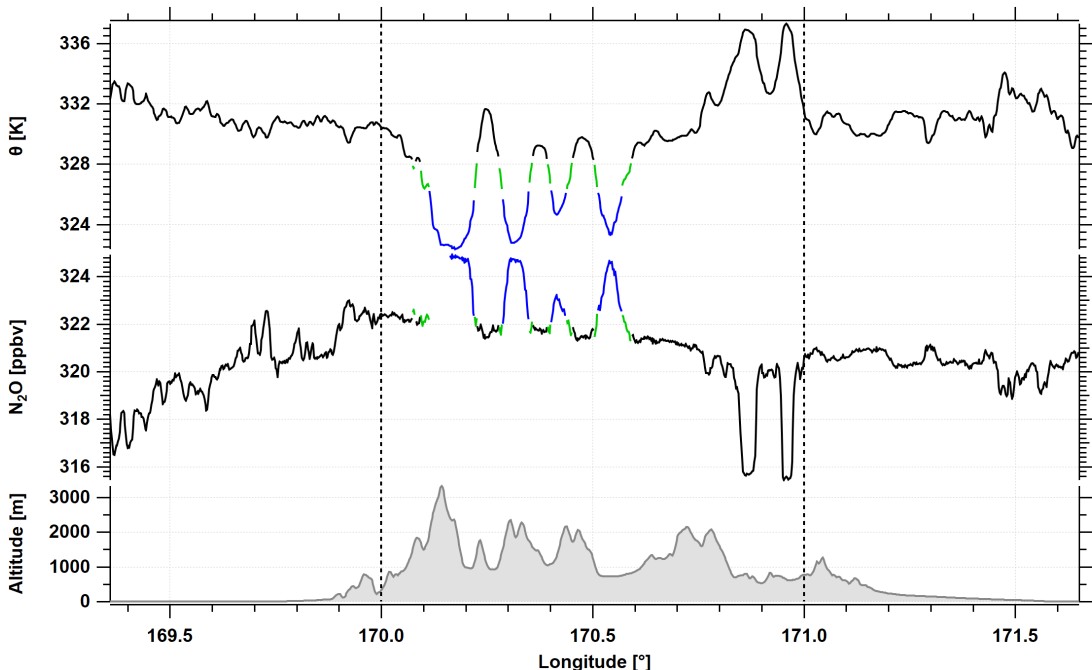

**Figure 7.** Time series of potential temperature $\Theta$ and $N_2O$ for the flight leg around 18:48-19:00 UTC just above the tropopause over the mountain. Colours indicate two different layers of air masses (black, blue) and a mixed layer inbetween (green) corresponding to Fig. **??**. Vertical lines mark the upstream ($<170°E$), above mountain ($170°E$-$171°E$), and downstream side ($>171°E$).

$\partial\chi/\partial\Theta = const$, Moustaoui et al. (2010)). In contrast, cross-isentropic (=diabatic) fluxes must change cross-isentropic gradients of species (i.e. $\partial\chi/\partial\Theta$) with respect to potential temperature. Therefore, a change of tracer slope (i.e. $N_2O$-gradients) as a function of $\Theta$ downwind the mountain is indicative for irreversible cross-isentropic tracer exchange which might have occurred above the mountain ridge. In a Lagrangian sense the occurrence of turbulence and turbulent mixing acts as a source

5 of tracer at a given isentrope, if the background tracer gradient changes with height in the inflow region (like e.g. at the tropopause).We therefore investigated if tracer gradients with respect to potential temperature $\Theta$ were changed due to the occurrence of gravity wave induced turbulent mixing by comparing local tracer profiles upstream and downstream the mountains ($\partial\chi/\partial\Theta|_{up} \neq \partial\chi/\partial\Theta|_{down}$). In particular, the gradient of the conservative tracer $N_2O$ at the tropopause is perfectly suited to test our hypothesis. Since $N_2O$ at the tropopause is not affected by chemistry it is purely under dynamical control. At the

10 tropopause, $N_2O$ exhibits a change of the vertical gradient and as a function of $\Theta$ (Fig. 7). Thus, mixing across the tropopause, e.g. via gravity wave induced turbulence may lead to a change of the $N_2O$ profile relative to $\Theta$.





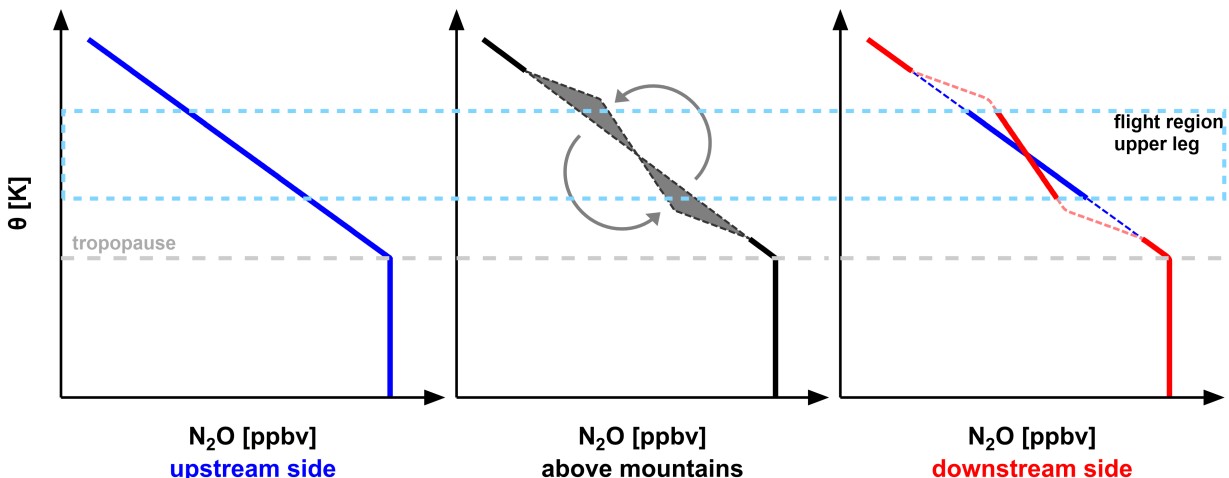

**Figure 8.** Schematic evolution of $N_2O$ versus potential temperature $\Theta$ in the presence of cross-isentropic mixing (indicated by the grey arrows) at the tropopause (e.g. by orographic wave-induced turbulence), which changes the $N_2O$ potential temperature gradient. The blue curve (left panel) shows the situation on the upstream side. The relation of $N_2O$ and $\Theta$ (blue) is modified by mixing (dotted black and grey shaded) over the mountains leading to a modified profile on the downstream side (red) compared to the original upstream relation (blue). The grey dashed line shows the tropopause and the light blue dash rectangle the flight region of the upper leg.

This is schematically shown in Fig. 8, which shows the evolution of the $N_2O$-$\Theta$ profile for a flow over a mountain assuming an effect of gravity wave induced cross-isentropic mixing on the $N_2O$ profile. The upstream side represents the unperturbed background $N_2O$ profile. Above the mountain orographic wave induced turbulence may occur which could potentially change the cross isentropic gradient of $N_2O$ relative to potential temperature $\partial N_2O/\partial\Theta$ at the tropopause. This effect of irreversible

5  mixing is schematically depicted by the gray shading in Fig. 8. Since the $N_2O$-gradient changes at the tropopause the upstream relation between $\Theta$ and $N_2O$ can be modified by turbulent mixing and the $N_2O$-$\Theta$ profile changes. As a result of the irreversible diabatic process the downstream side shows the modified profile of $N_2O$ versus $\Theta$ compared to the upstream region (red versus blue slopes).

Thus, in case of gravity wave induced turbulent mixing during flight FF09, we expect a steeper vertical $N_2O$-$\Theta$ gradient in the

10  inflow region upwind the mountains (i.e. a smaller $\Theta'$ $N_2O'$ ratio) than at the downstream side of the mountain ridge.





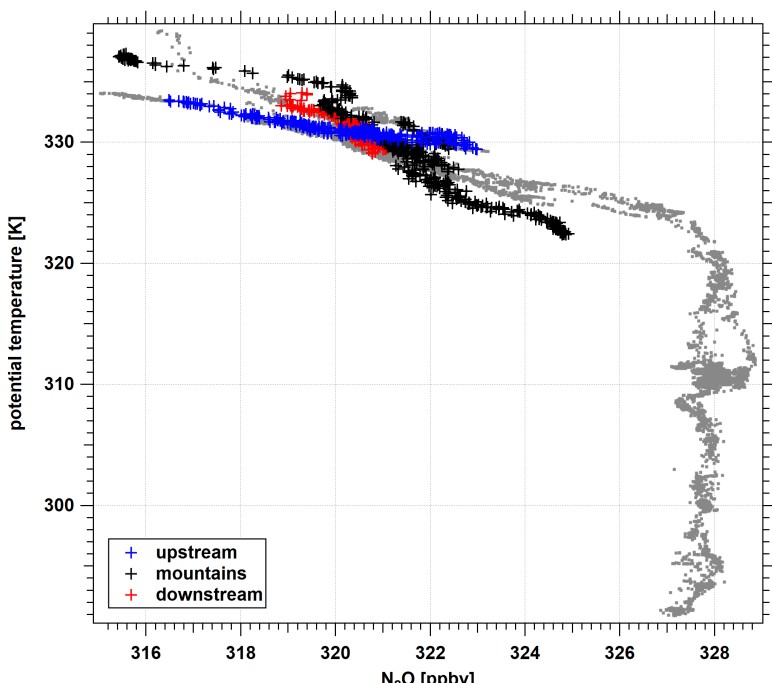

**Figure 9.** Profile of $N_2O$ during FF09 (gray) as a function of potential temperature $\Theta$. The upper leg of the southwestern part is shown with different colors, indicating different flight sections along the flight leg (blue: upstream, red: downstream, black: above the mountains, comp. Fig. 7).

Figure 9 shows the measured $N_2O$ profile as a function of potential temperature $\Theta$ for the entire flight FF09. The colored data points denote different longitudes relative to the mountain ridge to separate the inflow, across mountain, and downstream part of the flight leg between 170°E-171°E (see Fig. 7). As evident from Fig. 9 different slopes of $N_2O$ versus $\Theta$ appear on the upstream side, downstream side and above the mountains corresponding to the hypothesis described above. The $N_2O$-$\Theta$-

5  slope on the upstream side shows a strong decrease of $N_2O$ with potential temperature $\Theta$ and a compact relationship. On the downstream side the $N_2O$ decrease with respect to $\Theta$ is much weaker with intermediate values and larger variability above the mountains.





## 4.1 Scale analysis

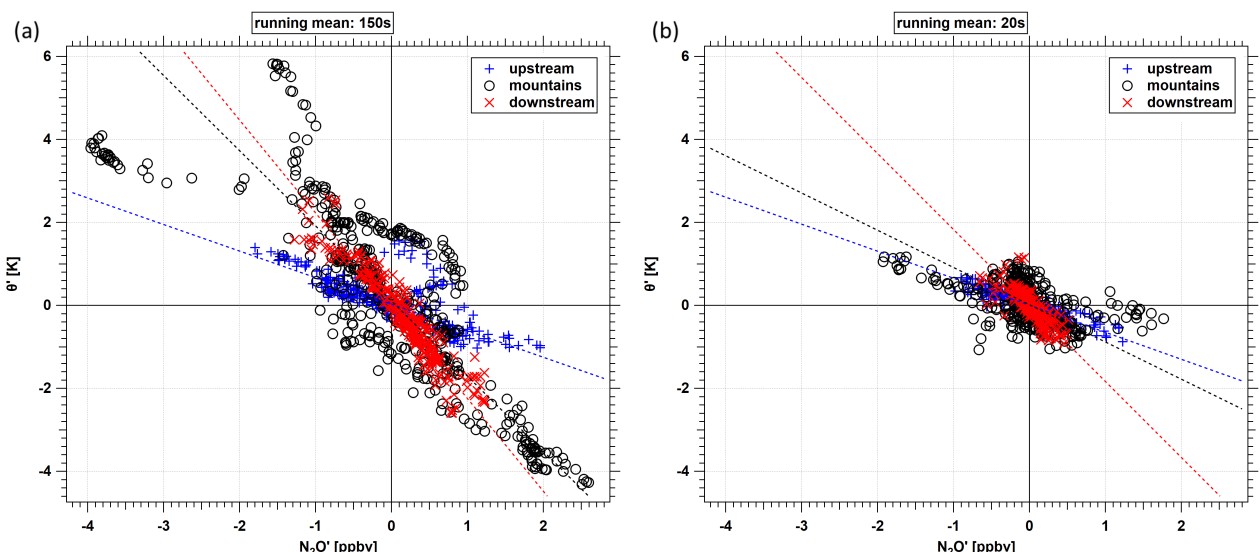

**Figure 10.** Relationship between $\Theta'$ and $N_2Oi'$ for upstream (blue), downstream region (red) and above mountains (black). Left: averaging period of 150 s. Right: averaging period of 20 s. The dotted lines indicate the slopes of the different flight segments. Note the changing slope over the mountains.

In a next step we performed a scale dependent correlation analysis as described below to further analyse the impact of the orographic waves on the $N_2O$-$\Theta$ relation and to account for a potential effect of different scales of the waves on cross-isentropic mixing. For this, we applied a Reynolds decomposition and separated the data into a mean part $\overline{\chi}$ and a perturbation part $\chi'$ (as well as $\Theta$ and $\Theta'$).

$$\chi(t) = \overline{\chi} + \chi'(t) \Leftrightarrow \chi'(t) = \chi(t) - \overline{\chi} \qquad (4)$$

The mean $\overline{\chi}$ is calculated with a boxcar average which works as a low-pass filter and removes high frequency variability. As given in detail further below, we analysed the data for different averaging periods to account for varying perturbation wave lengths and to analyse the effect at different scales.

$$\overline{\chi} = \frac{1}{t_2 - t_1} \int_{t_1}^{t_2} \chi(t)\, dt \qquad (5)$$

where $t_2 - t_1$ is the integration width.

$N_2O'$ and $\Theta'$ are linearly correlated with a particular slope defined by the upstream conditions. For linear non-dissipative waves the relation between $N_2O'$ relative to $\Theta'$ will therefore remain constant for different integration widths. In the presence of non-conservative dissipative processes the linear relation between $N_2O'$-$\Theta'$ and thus their slope will change as explained above.





As an example the effect of different integration intervals on the distribution of data points is shown in Fig. 10 for two different interval lengths. The applied fit accounts for errors in x and y direction (Press et al., 1987). The Figure shows the same subset of data using averaging periods on display of 150 s (Fig. 10 a) and 20 s (Fig. 10 b) corresponding to a horizontal scale of 33 km and 4 km, respectively. For wavelengths shorter than these dimensions the slopes at the downstream (red) and upstream 5 (blue) side only slightly change. Both however clearly show an increased slope at the downstream side (red) compared to the upstream (blue).

Above the mountains (black) the relation is perturbed, showing a high variability. The above mountain-slope for the long waves (i.e. averaging periods, left panel) is closer to the downstream side compared to the shorter wavelengths (right panel).

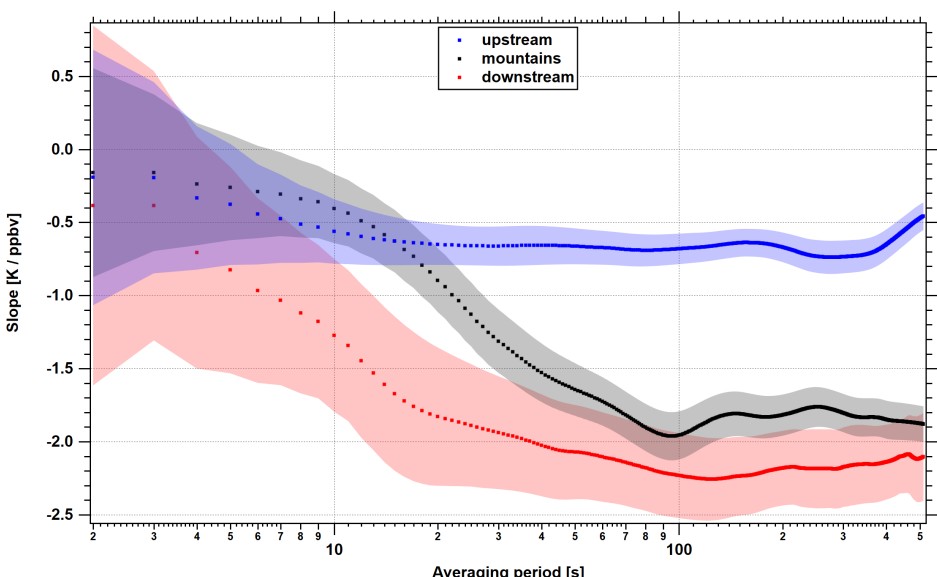

**Figure 11.** Scale dependent correlation anomaly analysis for different integration times showing the slope between $N_2O'$ and $\Theta'$ for different averaging periods (i.e. wavelengths) for upstream (blue), lee (red) and above mountains (black).

10     To account for different scales and thus increasing wavelengths, we subsequently increased the averaging period in steps of one second from 2 - 512 s. The corresponding development of the slopes between $N_2O'$ and $\Theta'$ is shown in Fig. 11. The blue curve is deduced from the data in the upstream region. With a value of about 0.6 K/ppbv the slope is almost constant over all averaging periods. This indicates that no cross-isentropic mixing perturbs the linear $N_2O'$-$\Theta'$ relationship at any wave period. Indeed in the long scale limit a clear separation for the downstream and above-mountain slope is evident. The black and red 15 curves show a slope change for different averaging periods. Above the mountains the slope starts to change at longer averaging periods (100 s) compared to the downstream side. The changes at the downstream side start at shorter averaging periods (40 s) indicating smaller scales or wavelengths relevant for the change of the $N_2O'$-$\Theta'$-relationship. Downstream and above-mountain slopes merge for longer periods to the new (modified) gradient with a slope around 2 K/ppbv.





Similar to Alexander and Pfister (1995) we used the upstream relation of $\Theta/N_2O$ to estimate the $N_2O$ amplitude which one would expect if only adiabatic transport by the gravity waves over the mountain would occur. The observed $\Theta$-amplitude of 8 K (Fig.7) would correspond to $N_2O=13$ ppbv, while only 4 ppbv are observed consistent with an impact of diabatic mixing processes changing the upstream relation.

5 In summary the slope changing behaviour would be consistent with a modification of the initial upstream $N_2O$-$\Theta$-profile across the mountains, where the relation between $N_2O$ and $\Theta$ is perturbed. When crossing the mountain ridge gravity wave induced turbulence affects the $N_2O$-$\Theta$-relation with persistent effect at the downstream side. This is evident from the different $N_2O'$-$\Theta'$ slope at larger wave lengths at the lee side compared to the upstream slope. Therefore we conclude that during FF09 mountain waves modified the slope $N_2O$-$\Theta$ at small scales and induced cross-isentropic turbulent mixing.

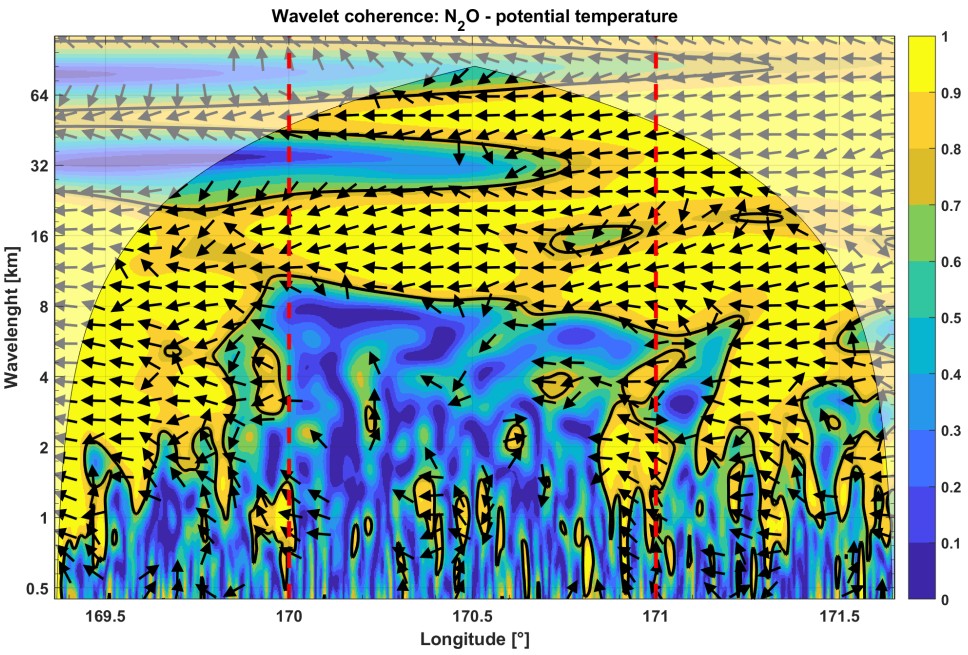

**Figure 12.** Wavelet coherence of $N_2O$ and potential temperature (wavelength = period · flight speed (216 m/s).) Arrows to the left indicate that $N_2O$ and potential temperature are shifted by $180°$. In regions with a coherence lower than 0.5 the phase indication is removed. The shaded region marks the cone of influence, where edge effects affect the analysis. The solid lines shows the 5% significance level. Yellow colors indicate a high coherence and blue colors indicate a low coherence. Vertical lines mark the upstream, above mountain, and downstream side.

To identify the leading spatial and temporal scales for the cross-isentropic (i.e. irreversible) mixing of $N_2O$ we analyzed the wavelet coherence between $N_2O$ and $\Theta$. Coherence is a measure of the intensity of the covariance of two time series. At the upstream side (< 170°E) there is mostly high coherence according to the fact that $\Theta$ and $N_2O$ co-vary across different time scales from 8-80 s. Further, the phase relation is constant at $180°$, which one would expect for a decreasing vertical $N_2O$ gra-





dient in the stratosphere, but increasing $\Theta$. Thus, both findings confirm the conclusion from the previous upwind slope analysis (Fig. 11, upstream side).

Above the mountains (from 170°E to 171°E) there is low coherence with values lower than 0.7 for time scales < 40 s accompanied by a breakdown of the phase relation of $N_2O$ and $\Theta$, both indicating a decrease of the covariance. On the downstream

5 side (from 171°E) especially at small periods higher coherence values and defined phase transitions appear compared to the above-mountain regime, albeit more variable than at the upstream side.

This matches roughly the results seen in Fig. 11. In upwind regimes with a high coherence $N_2O$ and the potential temperature $\Theta$ co-vary. The phase relation between them remains constant across scales and the calculated slope is unchanged too. Above the mountains the phase relation breaks down due to cross-isentropic mixing especially for wave periods smaller than 30 s.

10 Downwind a new slope relation reestablishes as a result of mixing above the mountain ridge, but with a defined phase relation again, but different slopes.

We therefore conclude that for wave periods with low coherence and the breakdown of the phase above the mountains the relationship between $N_2O$ and $\Theta$ is the result of gravity wave induced mixing leading to the observed $N_2O$-$\Theta$ slope change at the downstream side, where a modified slope relation establishes.

## 4.2 Fluxes

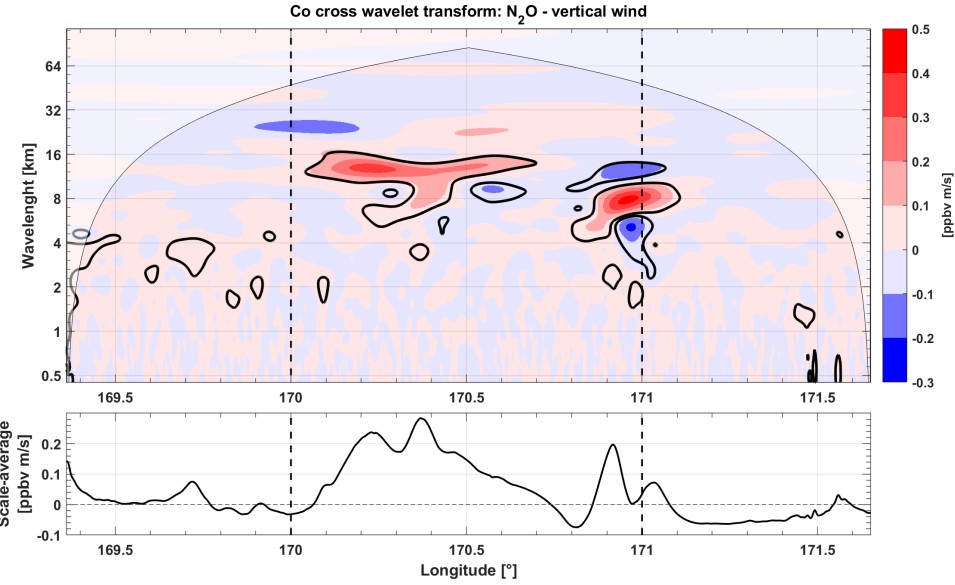

**Figure 13.** Top: Co-spectrum of the cross wavelet transformation of $N_2O$ and vertical wind. The shaded region marks the cone of influence in which edge effects play a role. The black contour lines indicate 5% significance levels against red noise. Red colors denotes a positive flux and blue colors indicate a negative flux. Bottom: Scale-averaged wavelet co-spectrum over all periods.





To estimate quantitative tracer fluxes we use the co-spectrum of the cross wavelet transformation between the vertical wind $w$ and $N_2O$ (see section 2.4). Unlike other methods the wavelet analysis has the advantage to resolve wave induced processes in space and time.

The wavelet co-spectrum is calculated from the real part of the cross wavelet transformation and gives the spectral contribu-

tions of vertical fluxes (Mauder et al., 2007). Fig. 13 shows the co-spectrum of the cross wavelet transformation of $N_2O$ and vertical wind $w$ for the higher flight level during FF09. There are two regions with enhanced fluxes within the 5% significance level (black solid line), which are both located above the mountains (compare Fig. 3). The first region is between 170.0°E and 170.6°E, the second at 170.8°E and 171.0°E The first region shows mainly positive trace-gas fluxes with values up to 0.50 ppbv m s$^{-1}$ at periods ranging from 8-16 km corresponding to the vertical wind energy maximum around $\lambda_x = 10$ km

Fig. 5. The second region exhibits upward and downward fluxes at slightly shorter scales from about 3-16 km. Here the strongest fluxes have values from about -0.22 to +0.56 ppbv m s$^{-1}$. The fluxes in both regions are co-located to enhanced wave occurrence above the mountains, as seen in Fig. 7 and Fig. 3 .

Since ozone was measured with a temporal resolution of 10 s we did not directly determine ozone fluxes in the present case. To give an estimate of the associated ozone flux we can use the $N_2O$-$O_3$ correlation slope, which is about -20 pppv($O_3$)/ppbv($N_2O$)

for the southern hemisphere July (Hegglin and Shepherd, 2007). This results in a negative flux of $O_3$ of 10 ppbv m s$^{-1}$ for the positive $N_2O$-fluxes.

### 4.3 Turbulence occurrence

The power spectral density (PSD) for the $\Theta$, and the vertical wind component $w$ in Fig. 14 shows a slope of -3 for frequencies smaller than 0.3 Hz, which is indicative for geostrophic turbulence (Zhang et al., 2015). For higher frequencies (i.e. shorter

wavelengths) the increase of the power spectral density at 0.5 Hz would be consistent with a potential source of turbulent energy above the mountains. Notably, a similar peak of the PSD of $w$ was observed during the START08 campaign for some flight sections in regions of turbulence occurrence over the Rocky Mountains (Zhang et al., 2015). However, the peak of the PSD from the vertical wind component around 0.5 Hz could also correspond to oscillations caused by the autopilot of the aircraft (Schumann, 2019). They report oscillations with wavelengths of 6.7 km in regions of high turbulence, which is factor

of 10 longer than in our case. Thus, an artificial non-atmospheric origin of the peak cannot be completely ruled out.

The slope of the PSD of both, $w$ and $\Theta$ turns towards -5/3 for frequencies exceeding 2 Hz, which can be related to isotropic turbulence. Fig. 14 also shows the PSD of $N_2O$ which show a slope of -5/3 for frequencies smaller 0.3 Hz and a -5/3 slope for higher frequencies. The transition of geostrophic to isotropic turbulence as indicated by the transition of PSD-slopes occurs in the frequency range, where the PSD of the vertical wind indicates a source of turbulent energy. Notably the PSD of $N_2O$

indicates a turbulent behaviour for high frequencies and thus the occurrence of turbulent fluxes corresponding to the analysis in the previous section.





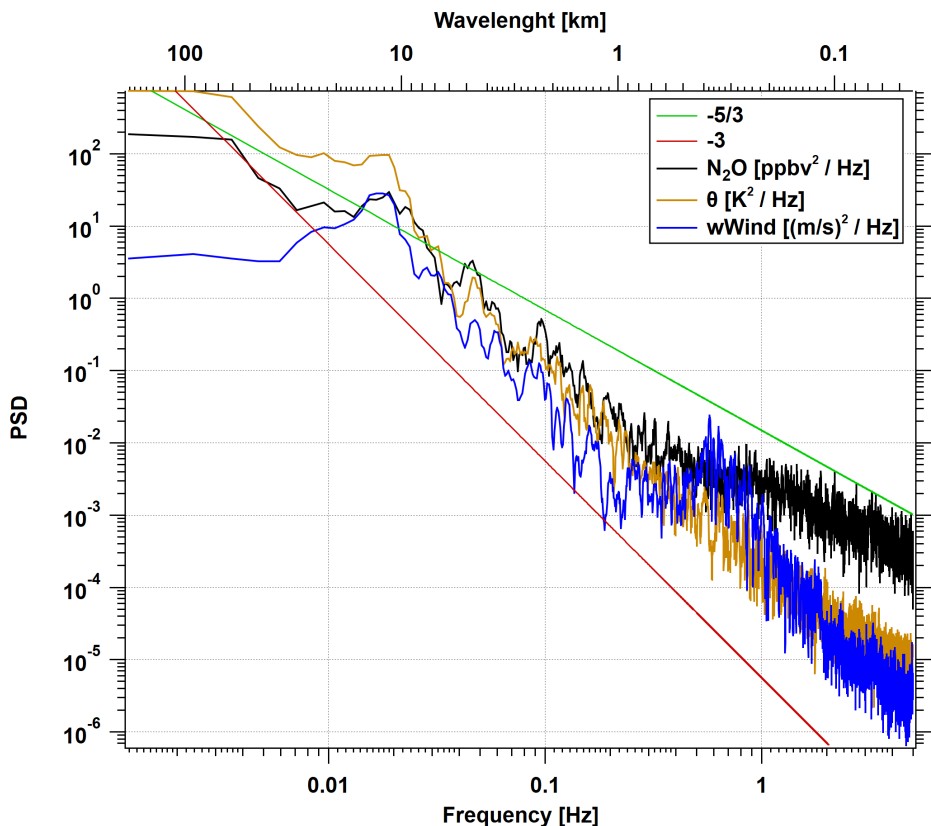

**Figure 14.** Smoothed power spectral density of vertical wind (blue), potential temperature (orange) and $N_2O$ (black) for the flight leg above the mountains. The green and red reference lines have slopes of -5/3 and -3.





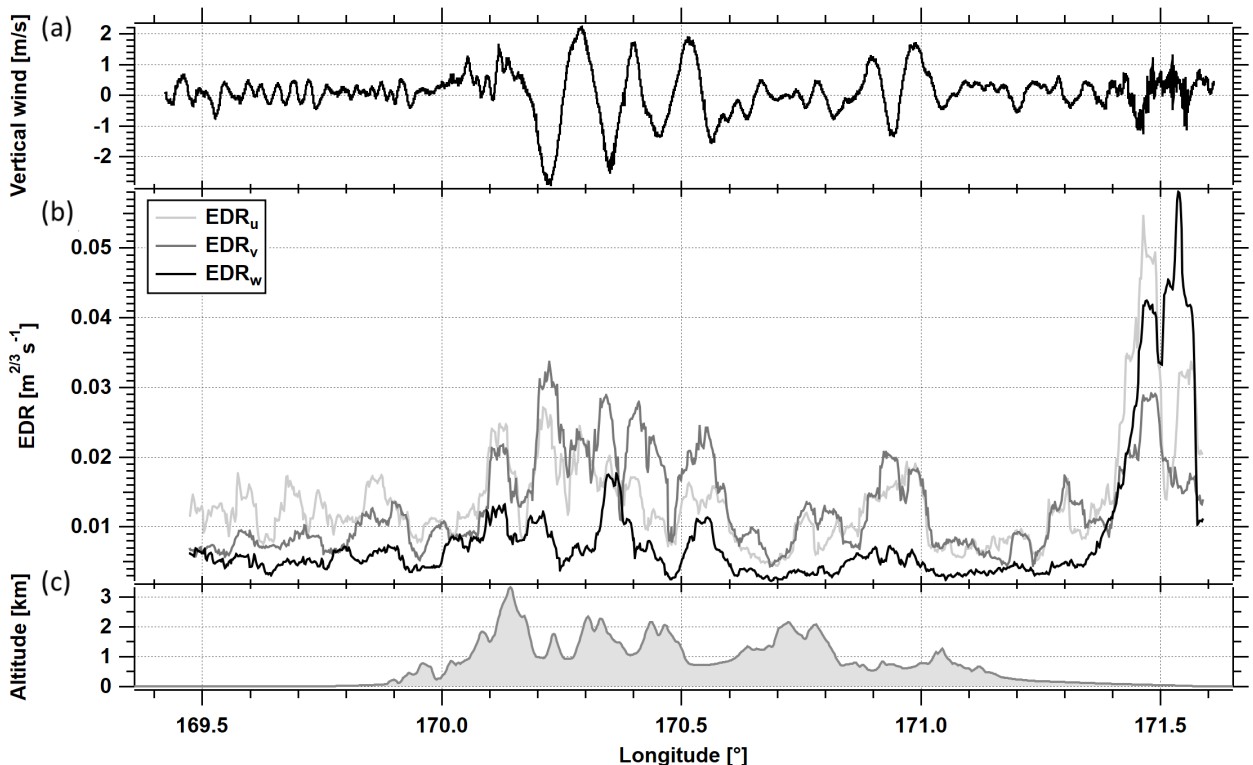

**Figure 15.** Timeseries of a) vertical wind, b) EDR for the measured wind components for the upper flight leg indicating weak, but non-vanishing turbulence during the time of flight above the Southern Alps (ororaphy shown in c).

Further support for our hypothesis and our results comes from the analysis of the cubic root of the eddy dissipation rate EDR=$\epsilon^{1/3}$ from the measured 3D-winds. For this analysis the we used the method by Bramberger et al. (2018) to calculate the EDR for the three wind components as measured by the aircraft (Fig. 15). Above the mountains the oscillations of the vertical wind velocity $w$ indicate the region of mountain wave occurrence. The EDR over the mountains appears to be weak

5 below the threshold of light turbulence of about 0.05 m$^{2/3}$s$^{-1}$ (Bramberger et al., 2018). However, the values of EDR$_{u,v}$ for the horizontal wind components over the mountains are similar to those of the end of the leg, when also EDR$_w$ was enhanced in the lee of the mountains.

Further support for our hypothesis and our results comes from the analysis of the occurrence of mountain wave induced turbulence using the GTG (Graphical Turbulence Guidance) using ECMWF operational analysis data (Bramberger et al., 2018;

10 Sharman et al., 2006; Sharman and Pearson, 2017). The GTG analysis matches the observed locations of wave occurrence during the flight and the regions of strong variability of the vertical wind (not shown). Though the values are too high this supports the conclusion that turbulence occurred in the region of the mixing events either shortly before or during the measurements.

The weak EDR at the upper flight in accordance with the weak turbulence occurrence as opposed to the lower leg (see Fif. 5) may be explained by the time difference between the two legs. As evident from the wave analysis orographic waves were





crossed during the first leg and led to wave breaking and momentum deposition in the region between the legs. At the higher leg only a weak turbulence signal remained one hour later when the second crossing took place. The change of the $N_2O$-$\Theta$ gradient is a unique indication of a diabatic process, which must have changed the gradient from the upstream to the downwind side exactly above the mountains. The evidence for strong orographic wave activity the lower level and propagating to the 10.9

km level serves as the only plausible explanation for this observations. The fact that at the higher level the turbulence is weak during the time of flight must be attributed to the time shift between the two flight legs and the high intermittency of turbulence occurrence.

## 5  Conclusions

We present an analysis of high resolution $N_2O$ measurements in the region of orographic gravity wave occurrence over the

Southern Alps in New Zealand during the DEEPWAVE 2014 campaign. These led to diabatic trace gas fluxes and a persistent local effect on the composition downwind the mountain and the above the tropopause.

The spectral analysis of the wind components measured along two vertically stacked levels indicates dissipation of momentum by orographic waves above the mountains between these levels and the generation of turbulence. The spectral energy of the vertical wind component shows strong signals at short horizontal wave lengths (<1 km) at the lower flight leg at 7.9 km. At the

higher leg, which was flown 75 min later in the stratosphere, horizontal wave lengths of 10 km dominate the energy spectrum of $w'$ with much weaker contribution at the shorter scale. Corresponding to the fluctuations of the vertical wind and potential temperature $\Theta$ also strong fluctuations of the tracer $N_2O$ were observed at the upper flight leg in the region of the occurrence of orographic waves. Based on the analysis of the $CO$-$N_2O$ relationship we could identify mixing between two layers of different air masses in the tropopause region. Upstream and downstream of the mountain different vertical gradients of $N_2O$ versus po-

tential temperature $\Theta$ were observed and enhanced variability of this gradient above the mountains. Since $N_2O$ is chemically inert a change of the $N_2O$-$\Theta$ relation must be due to cross-isentropic mixing effects.

A scale dependent slope analysis shows that mixing was initiated over the mountain ridge showing reversible displacements of tracer isopleths and $\Theta$. Above the mountains these fluctuations perturbed the compact slope of $N_2O$ in the inflow region around the tropopause.

The behaviour is also consistent with the indication for wave breaking and momentum deposition above the mountains between the two flight legs. Noting that the stratospheric flight leg was flown 75 min after the lower leg explains why the turbulent kinetic energy for $w'$ at short horizontal wave lengths is rather small compared to the lower leg. Still the power spectral energy spectra of $N_2O$ and $\Theta$ with slopes of -5/3 at the smallest scales can be seen as the result of the turbulence occurring potentially previously on this level.

The tracer conserves the effect of the highly transient turbulence occurrence. At the downstream side a modified compact $N_2O$-$\Theta$ relation establishes as a result of the wave induced turbulence above the mountains modulating the reversible air mass displacements induced by the gravity waves similar as described in (Moustaoui et al., 2010; Mahalov et al., 2011). This behaviour is confirmed by cross-wavelet analyses showing a breakdown of the coherence and phase relationship of $N_2O$ and $\Theta$



over the mountains.

The vertical fluxes of $N_2O$ are estimated to 0.5 ppbv m s$^{-1}$ corresponding to negative fluxes of $O_3$ of approximately 10 ppbv m s$^{-1}$. The change of the $N_2O$-$\Theta$-relationship from the upstream to the downstream side over the mountain ridge is a unique indicator for cross-isentropic (i.e. irreversible) turbulent exchange of species, which was initiated by the orographic waves. The fact, that

the modified relationship prevails downstream of the mountain shows, that the turbulence associated with the orographic waves was associated with cross-isentropic mixing.

This approach notably differs from local covariance analysis of vertical winds and tracers since it shows, that at least part of the kinematic fluxes contributed to a cross-isentropic component.

Diabatic trace gas fluxes are key for understanding the effect of mixing processes on the large scale composition of the

tropopause region and the lower stratosphere where they contribute to the mixing induced uncertainty of radiative forcing estimates (Riese et al., 2012). Though the occurrence of orographic waves has strong seasonality and high degree of transience (Fritts and Alexander, 2003; Rapp et al., 2021) regions of gravity wave activity are hotspots for turbulence occurrence at the tropopause (Alexander and Grimsdell, 2013; Fritts and Alexander, 2003). Our data show that this can have a persistent effect on the distribution of species and thus a potential forcing impact of radiatively active tracers by changing their isentropic

gradients. By subsequent isentropic transport as part of the stratospheric flow their impact has a strong non-local component downwind contributing to the overall mixing induced uncertainty of radiative forcing (Riese et al., 2012).

*Data availability.* ECMWF analysis data have been retrieved from the ECMWF MARS server. The airborne measurement data from the DEEPWAVE campaign are available through the HALO database (https://halo-db.pa.op.dlr.de/, last access: 6 June 2022).

*Author contributions.* PH conceived the study. HCL performed most of the data analysis with guidance from PH and DK. AD performed the gravity wave analysis. MB provided the EDR analysis. SM and PR performed the UMAQS measurements during DEEPWAVE and contributed to the flight planning. TK contributed to the turbulence analysis. AG provided the wind and turbulence data. MR and AD designed the DEEPWAVE campaign and the scientific flight planning. HCL and PH wrote the paper with contributions from DK; all authors contributed to the discussion of the results and the manuscript.

*Competing interests.* The authors declare that they have no competing interests.

*Acknowledgements.* The project was funded by the Deutsche Forschungsgemeinschaft (DFG) under grant HO 4225/6-1. The authors acknowledge support by the German Aerospace Center (DLR) Oberpfaffenhofen. The authors gratefully acknowledge support by the SFB/TR-301 (TPChange: The Tropopause Region in a changing atmosphere, Project-ID 428312742). Part of this research was funded by the German





research initiative "Role of the Middle Atmosphere in Climate" (ROMIC/01LG1206A) funded by the German Federal Ministry of Education and Research in the project "Investigation of the Life Cycle of Gravity Waves" (GW-LCYCLE), and by the Deutsche Forschungsgemeinschaft (DFG) via the Project MSGWaves (GW-TP/DO 1020/9-1 and PACOG/RA 1400/6-1).





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
