# Peer review of "Gravity wave induced cross-isentropic mixing: A DEEPWAVE case study"

_Atmospheric Chemistry and Physics, 2022_

## Author Comment (AC1)

**Overall Remarks:**

This paper presents a thorough and detailed analysis of a gravity wave induced atmospheric mixing event measured during the 2014 DEEPWAVE campaign in New Zealand. Through a combination of in-situ aircraft observations and ERA5 reanalysis data, the authors identify two distinct layers in the lower stratosphere with independent composition and isentropic character-

5 istics. They then show how the N2O-to-potential temperature gradient weakens due to gravity wave activity, and they identify signs of turbulence and trace gas fluxes to diagnose mixing between these two layers induced by the gravity waves. This mixing mechanism is distinct from past gravity wave-induced mixing studies in that it is cross-isentropic/diabatic/irreversible and yields nonlocal consequences downstream of the orographic mixing source.

10 Overall, this paper presents a clear and logical sequence of results and diagnostics supporting the main arguments of the text. I recommend that this paper be accepted for publication in ACP after addressing the minor revisions detailed below in two general comments on the use of terminology/writing structure and in line-by-line specific comments. The technical nature of the paper and use of complex, codependent sentence structures can make the arguments of the paper difficult to follow and less approachable to members of the larger gravity wave community. To enhance readability and make the paper more broadly

15 accessible to general audiences, a few simple modifications to the writing style and sentence structure would be beneficial as detailed below. There are also several specific comments regarding how variables are discussed/plotted and the possibility of using additional data from the HIAPER aircraft, if available.

**Authors response:**

20 We thank the reviewer for the careful reading and the constructive suggestions to improve the accessibility to a broader community. We tried to follow most of the suggestions and hope to have satisfied the criticism.

The major changes are a s follows:

1) We included a definition section to the manuscript to clarify the terminology. We checked the manuscript for a consistent wording particularly of the multi-word expressions.

25 We included a definition paragraph to the introduction as given below and adopted the text accordingly

2) We removed Fig. 1 and included the wind information into the former Fig. 2 as suggested.

3) We checked the terminology of the $N_2O$-$\Theta$ relationship and the expressions referring to slopes and ratios. We thereby kept our original idea to just use one way of expression to quantify slopes and ratios. This is directly deduced from the intuitively native way of analyzing vertical profiles with potential temperature as y-axis (similar to the discussion of temperature profiles,

30 which are commonly shown as temperature on the x-axis and $\Theta$ or altitude as y-axis). We therefore wanted to keep the emerging quotient throughout the manuscript with $N_2O$ in the enumerator and $\Theta$ in the nominator. We think this facilitates to follow any discussions instead of introducing inverted relationships. This is consistently done through the paper, independent of the analysis which relates $N_2O$ to $\Theta$. The reader does not need to link the ratio to a specific analysis step or Figure. We think, this facilitates the thinking. We added a note on this, when first introducing the scheme in former Fig. 8 *(former Fig. 9)*.

35 New text:

Note that we will use in the following the inverse relation $\partial\Theta/\partial N_2O$ to be mathematically be consistent with the profiles shown in Fig.8 *(former Fig. 9)* and in Fig. 7 *(former Fig. 8)*. We will consequently apply this convention with $\Theta$ in the numerator and $N_2O$ in the enumerator throughout the paper and will keep the same convention for any analysis which follows below.

40

**General Comments:**

1. The use of overlapping terminology to describe related transport phenomena, while technically correct in all instances, makes certain aspects of this manuscript esoteric and difficult to approach for readers lacking a comprehensive background in atmospheric chemistry and tracer transport (example: phrases describing cross-isentropic/diabatic/irreversible circulation/transport/

45 fluxes/mixing use pairs of these words somewhat interchangeably). The terminology in this manuscript also employs a number of related words with opposite meanings (example: a cross-isentropic process is not an isentropic process), which can confuse the reader when neither term is defined. When combined, these two terminology complexities make this paper less accessible for general audiences in the broader atmospheric community.

I suggest two terminology approaches to improve the readability and accessibility of the text:

a) Provide some basic definitions of terms when they are introduced to explain what they mean in the context of the other terminology used in the text (as an example, it is not explicitly stated until Page 13 that "cross-isentropic" and "diabatic" are used equivalently throughout the text because transport processes crossing lines of constant potential temperature, i.e. isentropes, are inherently diabatic). If the text states early on that cross-isentropic processes are both diabatic and irreversible, later descriptions in the text using "diabatic" and "reversible" can in many instances use the expression "cross-isentropic" since the reader will know this always refers to diabatic, irreversible processes. Though the text does define some terms like orographic gravity waves (Page 1 line 1) and passive tracers (Page 1 line 32), more definitions could be used throughout the text.

b) For multi-word dynamical behaviors, try to use consistent wording and word order to avoid confusing the reader. As an example, three sets of similar expressions are used on page 2 that alter the wording/order of two expressions meaning the same thing:

cross-isentropic mixing (line 8)
non-isentropic transport (lines 14-15)

vertical turbulent tracer flux (line 28)
turbulent vertical tracer flux (lines 29-30, word order switched)

Mountain wave induced tracer fluxes (line 29)
gravity wave induced vertical cross-isentropic tracer transport (line 31)

It may also be useful to employ acronyms for commonly used phrases to avoid having 8-word expressions for a physical concept like "gravity wave induced vertical cross-isentropic tracer transport". This will make it easier for the reader group multiword dynamical descriptions and parse out the surrounding sentence structure.

**Authors response:**
We took the suggestion and included a definition paragraph to the introduction to make the text more consistent. Some of the terms are redundant in aspects of their meaning (e.g. diabatic, irreversible, cross isentropic: these three all indicate a change of entropy and thus irreversibility of a process. Though they are used in a different way in the different communities. Cross-isentropic flux emphasizes the transport nature for irreversible transport of tracers and the quasi-vertical direction as opposed to quasi-isentropic mixing. All are diabatic and irreversible since $\Theta$ is not conserved indicating a change of entropy. With regard to tracer mixing there is also a mixture of terms and meanings between the communities - in dynamics mostly the dynamical processes are referred to by 'mixing', other communities refer to the aspect of irreversible constituent exchange by using the term 'mixing'.
We hope that we made the paper more clear with the newly included definition part. We are hesitant to include newly defined abbreviations since they make the text more difficult to fluent reading, if the reader has to look up non-common acronyms. Instead we followed the reviewers suggestion by avoiding swapping adjectives in multiword expressions and reducing their number.

2. Many sentences start with a pronoun (this/that/they/those/these, etc.) or a broad, unspecific term (our hypothesis, our conclusions, etc.) referring to the content of a previous sentence or paragraph. Often, due to the complexity of the referenced sentences/paragraphs, it is not clear what content these expressions refer to, requiring the reader to often go back to the referenced sentence to identify which topic from the previous sentence matches the description in the next sentence. To add clarity to the text, please try to avoid this sentence structure and instead state explicitly the topic of each sentence and the content being referenced. This can be applied throughout the text, with several examples identified in the Specific Comments below.

**Authors response:**
We checked the manuscript and added specific expressions instead of general wordings.

**Specific Comments line-by-line:**

———————————————

**Comment Page 1 Line 1:** please explain the term cross-isentropic when it is first introduced, clarifying how it refers to an
10 irreversible diabatic process to avoid confusion when these terms are later used to describe this same phenomenon.

**Reply to comment:** We introduced the term 'cross-isentropic' to emphasize the aspect of tracer mixing processes across
isentropes and to differentiate from the term quasi-isentropic mixing, related to stirring and mixing initiated mostly by plane-
tary waves (e.g. Plumb, 2002). Cross-isentropic by definition is a diabatic and importantly irreversible process, which relates
15 to our key message with regard to gravity waves.

———————————————

**Comment Page 1 Line 5:** remove the comma after "shows"

20 **Reply to comment:** We removed it.

**Changes in manuscript:** A detailed analysis of the observed wind components shows, that both flight legs were affected
by vertically propagating gravity waves with momentum deposition and energy dissipation between the two legs.

25 ———————————————
**Comment Page 1 Line 8:** Clarify the quantity of the referenced tracer gradient (I believe you refer to a cross-isentropic gradi-
ent of tracer concentration, but this isn't specified)

**Reply to comment:** We clarified it accordingly.
30
**Changes in manuscript:** For the stratospheric data we identified mixing leading to a change of the cross-isentropic tracer
gradient of $N_2O$ from the upstream to the downstream region of the Southern Alps.

———————————————
35 **Comment Page 1 Line 10:** please define theta as potential temperature when the variable is first used

**Reply to comment:** Changed as suggested.

**Changes in manuscript:** Based on the quasi-inert tracer $N_2O$ we identified two distinct layers in the stratosphere we identified
40 two distinct layers in the stratosphere with different chemical composition on different isentropes as given by constant potential
temperature $\Theta$.

———————————————
**Comment Page 1 Line 18:** comma after "N2O"

45
**Reply to comment:** We added it.

**Changes in manuscript:** The $N_2O$-$\Theta$-relation downwind the Alps modified by the gravity wave activity provides clear ev-
idence that trace gas fluxes, which were deduced from wavelet co-spectra of vertical wind and $N_2O$, are at least in part

cross-isentropic.
* * *
**Comment Page 1 Line 22:** clarify that these "irreversible diabatic" trace gas fluxes are cross-isentropic to be consistent with the terminology introduced in line 1 and used throughout.

**Reply to comment:** We clarified it.

**Comment Page 1 Line 23:** Define UTLS in its first use in the text

**Reply to comment:** We changed it.

**Changes in manuscript:** This finally leads to irreversible  (i.e. diabatic) trace gas fluxes across isentropes and thus has a persistent effect on the  upper troposphere/lower stratosphere (UTLS) trace gas composition.
* * *
**Comment Page 2 Lines 8, 14-15, 28-31:** See General Comment 1 regarding consistent terminology and word order

**Reply to comment:** According to the general comment above we added an explanation, which will provide our use of the terms cross-isentropic, diabatic and mixing and our intention of their use.

**Changes in manuscript:** We added the following sentence: We will use the term 'cross-isentropic' to emphasize the irreversible (entropy changing and therefore diabatic) nature of this process. Further the term 'cross-isentropic' allows to distinguish from 'quasi-isentropic mixing'. The latter is driven by synoptic and planetary waves leading to stirring and mixing best approximated along isentropes.
Diabatic processes lead to an irreversible redistribution of tracers, which must be therefore cross-isentropic providing a tracer flux crossing isentropes.
We modified the sentence: Direct observations of gravity wave induced  cross-isentropic  mixing are sparse, since
* * *
**Comment Page 2 Line 6:** Define UTLS in abstract on page 1, in which case the definition is not needed here

**Reply to comment:** We adjusted it.

**Changes in manuscript:** However, in the  UTLS observations of gravity waves from aircraft and balloon soundings are essential for process studies beyond the resolution of satellites
* * *
**Comment Page 2 Line 8:** Change "They" to "Gravity Waves". Due to complexity of general sentence structure, the manuscript will be clearer if sentences that start with a pronoun (it/this/that/these/those) referring to something from a previous sentence are changed to instead state the referenced topic from the previous sentence/paragraph explicitly.

**Reply to comment:** Changed as suggested.

**Changes in manuscript:**  Gravity Waves propagate across the UTLS where static stability increases at the tropopause
* * *
**Comment Page 2 Line 11:** Change "Both" to "Both types of instabilities" for clarity - see previous comment.

**Reply to comment:** Changed as suggested.

**Changes in manuscript:**  Both types of instabilities may lead to the occurrence of turbulence, particularly when wave
breaking occurs with potential subsequent mixing of trace species
* * *
**Comment Page 2 Line 15:** Comma after "barrier"

**Reply to comment:** We added it.

**Changes in manuscript:** The tropopause as a central feature of the UTLS acts as a dynamical barrier, for transport of species
and the formation of trace gas gradients at the tropopause
* * *
**Comment Page 2 Lines 15, 17 and 18:** clarify the text to make it clear that "cross-isentropic mixing" (17) and "irreversible
trace gas exchange" (18) are the required diabatic processes referred to in line 15.

**Reply to comment:**
We added a clarification to the manuscript. See comment p.2 l.8, l.14-15, l.28-31 above
* * *
**Comment Page 2 Line 16:** commas after "addition" and "occurrence"

**Reply to comment:** We added the commas.

**Changes in manuscript:** In addition, turbulence occurrence, associated with wind shear above the tropopause
* * *
**Comments Page 2 Line 25:** comma after "fold"

**Comments Page 2 Line 25:** remove "occurrence"

**Reply to comment:** Changed as suggested.

**Changes in manuscript:** Based on airborne observations in a tropopause fold, Shapiro (1980) identified ozone and parti-
cle fluxes in regions of turbulence  and shear.
* * *
**Comment Page 3 Line 7:** Remove "steps in here to"

**Reply to comment:** Changed it accordingly.

**Changes in manuscript:** This study  provides evidence on the basis of observed passive tracers
* * *
**Comments Page 3 Line 8:** remove "will""

**Comments Page 3 Line 9:** remove "non-local", as it is already clear from the text that the location downwind of the turbulent mixing region is non-local to the turbulence.

**Comments Page 3 Line 9:** change "downwind" to "downwind of"

5  **Reply to comment:** Changed as suggested.

**Changes in manuscript:** We  investigate how orographic gravity wave induced turbulence leads to a  persistent effect on the UTLS composition downwind of the turbulent mixing region.

10  ___________________
**Comment Page 3 Line 23:** change "and covered" to "that covered"

**Comment Page 3 Line 23-24:** change "upper troposphere lower stratosphere" to UTLS

15  **Comment Page 3 Line 23:** change "providing" to "and provided" - 80 km altitudes are outside of the UTLS region.

**Reply to comment:** Changed as suggested.

**Changes in manuscript:** Airborne measurements were carried out from Christchurch during June and July 2014  that
20  covered the  UTLS  and provided remote sensing data up to 80 km.
* * *
**Comment Page 3 Line 26:** Was there a corresponding HIAPER flight for the Falcon flight for this case study on 12 July? Later statements in the text say the FALCON flight legs were too short to measure the longer gravity wave horizontal wavelength
25  and that two aircraft flying at close altitudes are required to calculate the flux divergence. Many of the coordinated flights in DEEPWAVE using both aircraft had HIAPER flying higher/longer legs near to where the FALCON was flying. Was this the case on 12 July, and if so, could these statements in the text be addressed by looking at HIAPER data from corresponding legs? If there was no corresponding HIAPER flight, please clarify this in the text and also state explicitly that all observations used for this flight are from instruments on the FALCON (and not HIAPER) aircraft-this is never stated in the text.
30
**Reply to comment:** There were no flights performed on 12.July with the HIAPER aircraft. The Falcon camppaign was framed by HIAPER on 11.July and 13.July. Further there was no trace gas payload installed on the HIAPER aircraft, which could be compared to our tracer data.

35  **Changes in manuscript:** The two aircraft partly performed coordinated flights in the tropopause region to study the propagation and potential dissipation of gravity waves in this region. During the 12. July, which is the day of the analysis in this paper no HIAPER flight was performed.
* * *
40  **Comment Page 3 Line 30:** change "2015)." to "2015) onboard the DLR Falcon." See previous comment.

**Reply to comment:** We changed it accordingly.

**Changes in manuscript:** Tracer measurements of $N_2O$ and CO were performed using the 'University MAinz Quantum Cas-
45  cade Absorption Spectrometer (UMAQS, Müller et al., 2015) onboard the DLR Falcon.
* * *
**Comment Page 4 Line 1:** change "CO" to "CO concentrations" to clarify what quantity this instrument measures for N2O and CO

**Reply to comment:** The instrument measures the absorption of Infrared radiation by the absorber density and thus a quantity scaling with the concentration. This concentration is converted to volume mixing ratios using pressure and temperature in the measurement cell. Unlike the concentration, the volume mixing ratio is conserved under pressure changes. Generally volume mixing ratios (or shortly referred to as mixing ratios) are reported for tracer transport studies given in parts per billion by volume (ppbv) corresponding to nanomole per mole in SI-units.

**Changes in manuscript:** The instrument is capable of simultaneously measuring the species $N_2O$ and CO reported here as volume mixing ratios in ppbv with a temporal resolution of 10 Hz
* * *
**Comment Page 4 Line 4:** Define sigma in this context-I believe it is the standard deviation in this case.

**Reply to comment:** Correct. We changed it accordingly.

**Changes in manuscript:** is on the order of 0.05 ppbv (1 standard deviation $\sigma$) for $N_2O$ and CO, respectively.
* * *
**Comment Section 2.3:** It is not always clear in your figures which data is from ECMWF and which data is from the aircraft - please distinguish your data sources in figures containing a mixture of model data and observational data.

**Reply to comment:** We modified the captions of the Figures accordingly to clarify this.

**Changes in manuscript:** We changed the respective captions (see Figs. 1, 2 *(former Figs. 2, 3)*).
* * *
**Comment Page 4 Line 21:** Is the "5% significance level" referenced in wavelet figure captions the same as the "95% confidence level" stated in the text? If so, please use consistent terminology or define the 5% significance level in the main body of the text.

**Reply to comment:** The significance at the 5% level is equivalent to the 95% confidence level

**Changes in manuscript:** To reveal periods with significant wavelet power we determined the 95% confidence level (which is equivalent to the 5% significant level) in the respective analyses below as described in Torrence and Compo (1998).
* * *
**Comment Page 4 Line 22:** To be consistent with your use of American English spellings of words such as "color" rather than "colour", use "analyze" in place of "analyse"

**Reply to comment:** The words vapour, analyse, colour, grey, behaviour are changed to American English.
* * *
**Comment Figure 1:** Figure 1 is not utilized in the text and may be unnecessary. The flightpath is shown already in Figure 2, and arrows could be added to indicate flight direction in that figure. The text discussion of the tropopause height also does not refer to Figure 1 - it only references the red line in Figure 2b on Page 6, and the discussion of the "approaching upper level trough" references Gisinger et al (2017) rather than Figure 1. Please provide more direct references that utilize Figure 1 to justify its inclusion in the text.

**Reply to comment:** We removed Figure 1 and included the wind information to former Figure 2.

─────────────────

**Comment Figure 2:** Figure 2 In panel b, consider adding gray shading of the flight sections that are later used for detailed analysis to make it easier to see which part of the ECMWF modeled wave response is sampled in the regions of interest in Figure 3.

**Reply to comment:** We added red boxes framing the respective regions.

**Changes in manuscript:** see added figure

[Figure]

─────────────────

**Comment Figure 2 Caption:** change "horizontal" to "ECMWF horizontal" to clarify the data source

**Comment Figure 2 Caption:** change both instances of "denotes" to "denote" - the subject ("lines") is plural in both cases.

**Reply to comment:** Changed as suggested.

**Changes in manuscript:** Divergence of the ECMWF horizontal wind during the time of flight a) at 250 hPa and b) as vertical cross section along the flight track indicating the signature of gravity waves over the Southern Alps. The solid red line denote the -2 pvu isoline, the black dashed lines  denote contours of the horizontal wind velocity (10, 15, 20, 25 m s$^{-1}$ in a) and 10, 20, 30 m s$^{-1}$ in b)) and the gray dashed lines in b)  denote contours of potential temperature.

─────────────────

**Comment Page 6 Line 8:** There is no panel (e) in Figure 2 - please clarify this reference.

**Reply to comment:** Figure 2e in Gisinger et al. 2017. Clarified in the text.

**Changes in manuscript:** According to Gisinger et al. (2017) the synoptic situation can be characterized by a trough located west of New Zealand with a weak surface low south of the Islands causing northwesterly winds in the troposphere (TNW regime,  Figure 2e in Gisinger et al. (2017)).

─────────────────

**Comment Page 6 Line 13:** change "South Island" to "the South Island"

**Reply to comment:** We changed it accordingly.

**Changes in manuscript:** These conditions led to the excitation of mountain waves and generated varying and moderate gravity wave responses over the South Island
* * *
**Comment Page 6 Line 13:** change "horizontal" to "ECMWF horizontal" to explicitly state the data source.

**Reply to comment:** Changed as suggested.

**Changes in manuscript:** Fig. 1a *(former Fig. 2a)* shows the divergence of the ECMWF horizontal wind at 250 hPa at 18:00 UTC
* * *
**Comment Figure 3 Caption:** Does analyzed PV come from ECMWF? If so, please state this explicitly in the caption.

**Comment Figure 3 Caption:** change "potential vorticity" to "potential vorticity (PV)" to link with figure labels.

**Comment Figure 3 Caption:** Clarify what quantity of N2O and CO is plotted. The units in the plot seem to indicate that these are concentrations, yet the text refers to the N2O line as the mixing ratio (line 10), making the quantity that is plotted in the figure ambiguous. See General Comments above regarding the use of consistent terminology.

**Reply to comment:** Correct, PV also comes from ECMWF. We clarified it.

**Changes in manuscript:** Time series of a) potential temperature $\Theta$ from the measurements (black), altitude (blue) above surface elevation (filled blue), b) vertical wind (black), horizontal wind (blue), c) N2O (blue) and CO (black) volume mixing ratios and d) ECMWF potential vorticity PV interpolated along the flight track.
* * *
**Comment Page 8 Line 2:** $\theta$ should be defined as potential temperature much earlier in the text, not here.

**Reply to comment:** Changed as suggested.

**Changes in manuscript:** The fluctuations of  $\Theta$ reached an amplitude
* * *
**Comment Page 8 Figure 4:** label the upper leg and lower leg panels on the right side of the plots

**Reply to comment:** We changed the Figure and added respective labels.

**Changes in manuscript:** see added figure
* * *
**Comment Figure 4 Caption:** State in the caption that the data plotted from the upper leg and lower leg corresponds to the shaded regions of Figure 3.

**Comment Figure 4 Caption:** From the text (Page 7 Line 17) and the tropopause height in Figure 2, the upper leg is "just above the tropopause", whereas the lower leg is farther from the tropopause and shouldn't be labeled as "just below the tropopause".

[Figure]

If anything, the clarifying statement in the figure caption should indicate that the upper leg is just above the tropopause, as in the text. See General Comments above regarding the use of consistent wording.

**Reply to comment:** Changed as suggested.

**Changes in manuscript:** Cross section of the two southern stacked flight legs crossing the Southern Alps (gray shaded regions of Fig. 2 *(former Fig. 3))* showing $N_2O$ (green), $\Theta$ (blue) and vertical wind $w$ (black) for the upper leg at 10.9 km (top three panels) and the lower leg at 7.9 km with surface elevation (bottom). Both legs are separated by 75 minutes in time. The upper leg lies in the lower stratosphere just above the tropopause, the lower leg lies in the upper troposphere .

─────────────────────

**Comment Page 9 Line 3:** remove ",which has a lifetime of 110 years in the lower stratosphere," - this lifetime information is restated later in the text where it is relevant to the discussion, but it is not important to state this information a second time in this location.

**Reply to comment:** We removed it.

**Changes in manuscript:** The passive tracer nitrous oxide $N_2O$ indicates corresponding fluctuations at the upper level in the stratosphere.

─────────────────────

**Comment Page 9 Line 7:** Consider replacing "such a breakdown of scales" with "such turbulence" to unambiguously refer to the "occurrence of turbulence" mentioned in the previous sentence. See General Comments above regarding the use of consistent wording.

**Reply to comment:** Replaced as suggested.

**Comment Page 9 Line 7:** Maybe state more clearly in the text that you identify a kinematic flux of N2O by collocated, phase-shifted fluctuations of theta and w indicating a nonzero w'theta' that has corresponding fluctuations in N2O concentrations.

**Reply to comment:** At this point we only want to indicate, that the time series indicate the potential for emerging kinematic fluxes without proving this at this point quantitatively.

**Changes in manuscript:** At the upper level such  turbulence is not prominent, although the fluctuations of $\Theta$, $w$ and $N_2O$ (Fig. 3 *(former Fig. 4)*) are indicative  of at least a potential kinematic flux of $N_2O$,  with only weakly pronounced small scale variability of $w'$.

———————————

**Comment Page 9 Lines 11-13:** Change word order to "The vertical turbulent kinetic energy was larger in the lower leg $(\overline{w'^2} = 0.70\,\text{m}^2\,\text{s}^{-2})$ than in the upper leg $(\overline{w'^2} = 0.53\,\text{m}^2\,\text{s}^{-2})$, where the overline denotes the average over the whole 200 km flight leg." This will make the sentence less confusing.

**Reply to comment:** We changed the sentence as suggested.

**Changes in manuscript:**  The vertical turbulent kinetic energy was larger in the lower leg $(\overline{w'^2} = 0.70\,\text{m}^2\,\text{s}^{-2})$ than in the upper leg $(\overline{w'^2} = 0.53\,\text{m}^2\,\text{s}^{-2})$, where the overline denotes the average over the whole 200 km flight leg.

———————————

**Comment Page 9 Line 14:** Does "this energy" refer to the energy in the lower leg or the energy in the upper leg? Please state explicitly which leg is referenced here. See General Comments above regarding unclear use of pronouns referring to previous sentences.

**Reply to comment:** "this energy" refers to the lower leg. We changed it accordingly.

**Changes in manuscript:** However,  the energy of the lower leg seems to reside in scales smaller than about 1 km

———————————

**Comment Page 10 Line 5:** Was there a corresponding HIAPER flight with longer legs that could identify the longer gravity wave horizontal wavelengths? Clarify earlier in the text whether both aircraft were flying, and if there is corresponding HIAPER data, perhaps it is worth checking to see if the longer wavelength can be identified.

**Reply to comment:** As mentioned in a previous comment above no HIAPER flights took place at this date.

———————————

**Comment Page 10 Line 10:** Reference Table 1 values in the text where you mention the zonal momentum fluxes

**Reply to comment:** True. Changed it accordingly.

**Changes in manuscript:** The specific zonal momentum fluxes $\overline{u'w'}$ (Tab. 1) are negative above the mountains

———————————

**Comment Page 10 Line 15:** I believe that the vertical derivative is taken by comparing values from the two flight legs at different altitudes, right? Or is the estimate from ECMWF? Perhaps clarify how this value is estimated - it is confusing to say you take a vertical derivative from flight legs that only sample horizontally unless more information is provided.

**Reply to comment:** The vertical derivative is taken by comparing values from the two flight legs.

**Changes in manuscript:** An estimate of the vertical momentum flux divergence based on the values from the stacked flight legs
* * *
**Comment Page 10 Line 19:** Clarify in the text that you are referring back to wind components that are plotted back in Figure 3 and/or Figure 4.

**Reply to comment:** Changed as suggested.

**Changes in manuscript:** This argument is supported by the small-scale signatures found in all wind components downstream of the coherent waves in the lower leg (see Fig. 2 and Fig. 3 *(former Fig. 3 and 4)*).
* * *
**Comment Figure 6:** The use of similar colors for different variables makes it more difficult to explain and distinguish which variables are plotted. It would be more effective to use different colors (instead of 3 shades of gray/black) and add a legend to the plot identifying each plotted variable color.

**Reply to comment:** For a better identifiability the data points from the lower flight leg are now in orange.

**Changes in manuscript:** see added figure

[Figure]
* * *
**Comment Figure 6 Caption:** It is unclear which datapoints are "colored data points" since all datapoints are colored. Does this sentence refer to all the datapoints in the figure or a specific subset?

**Reply to comment:** We changed it accordingly.

**Changes in manuscript:**  Black, blue and green data points denote the upper south-western flight leg from 18:48 UTC to 19:06 UTC
The lower leg lies entirely in the troposphere as indicated by the  orange data points of $N_2O$ = 328 ppbv.
* * *
**Comment Page 12 Lines 4-5:** replace "The orographic waves at the lower leg" with "N2O concentrations in the lower leg (black) to clarify that you're referring to the N2O concentration in the plot

**Reply to comment:** Replaced as suggested.

**Changes in manuscript:** The orographic waves at the lower leg appear at almost constant $N_2O$  volume mixing ratios of 328 ppbv.
* * *
**Comment Page 12 Line 11:** "N2O mixing ratios": Perhaps you should identify and show fits of these mixing ratios in the plot, as people from outside the field may not understand that you refer to regions of near-constant ratios between concentrations of N2O and CO as "mixing ratios" when the term is not defined or plotted explicitly. You could also clarify that "N2O mixing ratio" is the ratio between N2O and CO, otherwise it is unclear why you don't refer to it is the "CO mixing ratio" or the "N2O:CO mixing ratio".

**Reply to comment:** We thank the reviewer for this point since it reflects the importance of a exact language and wording to facilitate understanding across different communities. As stated above (and included now in the manuscript referring to the reviewer comment to p.4, l.1) we clarified the term "mixing ratio". Generally the term mixing ratio (or more correct "volume mixing ratio") refers to the abundance of a gaseous species by reporting its volume mole fraction (in nanomole per mole or ppbv). It has nothing to to do with a ratio of two species. This may indeed lead to confusion when comparing to different species. We therefore changed the wording throughout the manuscript.

**Changes in manuscript:** We changed the term "mixing ratio" to "volume mixing ratio" throughout the manuscript.
* * *
**Comment Page 12 Line 11:** your "detailed analysis" is not shown - please provide more information on how these two temperature ranges were identified and what their physical significance (if any) is.

**Reply to comment:** The corresponding analysis is linking former Fig. 6 and former Fig. 7. The time series of the potential temperature was colorized in former Fig. 7 corresponding to former Fig.6 to show the spatial and temporal distribution of data points, which allows to identify three layers

**Changes in manuscript:** A detailed analysis (see Fig. 7 *(former Fig. 6))* shows that the two branches of the correlation can be assigned to two distinct potential temperature intervals
* * *
**Comment Page 12 Line 13:** What is a "compact relation"?

**Reply to comment:** Compact is indeed a qualitative description describing the less scattered data populations with a high data density and small scatter variability. However, it is well known that different air masses in the stratosphere show distinct correlations between different tracers (e.g. Plumb, 2007; Hoor et al., 2002), which can be used to identify a mixture of composition of the respective air masses.

**Changes in manuscript:** Notably, the data points (marked in green) which fall between the two  data clouds ($N_2O$ < 324 ppbv) forming two compact branches
* * *
**Comment Page 12 Line 14:** Please explain how the "compact relations" are given above.

**Reply to comment:** In Fig. 6 *(former Fig. 7)* the $N_2O$-CO correlation has two compact branches in the stratosphere where the distribution of data points in the scatter plot show a small variability, i.e. the slope of each branch has high coefficient of determination.
* * *
**Comment Page 12 Line 17:** comma after "context"

**Reply to comment:** Added.

**Changes in manuscript:** the tracer-tracer data of the scatter plot in a geophysical and meteorological context, Fig. 6 *(former Fig. 7)* shows the time series of potential temperature
* * *
**Comment Page 12 Line 20:** change inbetween to "between", here and elsewhere in the text

**Reply to comment:** Changed here and elsewhere.

**Changes in manuscript:** Notably those points which indicate mixing in the tracer-tracer scatter plot fall between the distinct layers.
* * *
**Comment Page 12 Line 28:** change "vertically closely stacked levels" to "closely stacked vertical levels"

**Comment Page 12 Line 29:** change "can not" to "cannot"

**Comment Page 12 Line 29:** Clarify earlier in the text whether there was one or two aircraft flying on 12 July.

**Reply to comment:** Changed and clarified as suggested.

**Changes in manuscript:** However, this would require simultaneous measurements of the tracer of interest on two  closely stacked vertical levels, which  cannot be accomplished with one aircraft (as was the case here).
* * *
**Comment Page 12 Line 30:** change "km potential" to "km, the potential"

**Comment Page 12 Line 31:** change "levels" to "flight levels"

**Reply to comment:** Changed as suggested.

**Changes in manuscript:** However, due to the large vertical spacing of 3 km, the potential influence from large scale horizontal advection could strongly impact the flux divergence estimates between the two flight levels.
* * *
**Comment Figure 7 Caption:** fix the broken figure reference "Fig. ??"

**Reply to comment:** The figure reference was repaired.

**Changes in manuscript:** Colors indicate two different layers of air masses (black, blue) and a mixed layer between (green) corresponding to Fig.  5 *(former Fig. 6)*.
* * *
**Comment Page 13 Line 1:** It should be stated much earlier in the text that cross-isentropic fluxes are diabatic. 1 and onward: the text refers to species gradients as d(X)/d(theta), yet the plotted gradient in Figure 11 appears to be inverted as d(theta)/d(X). Because the text indicates that the tracer slope changes as a function of theta (instead of saying the theta slope changes as a function of the tracer), it would be much clearer to plot d(X)/d(theta) rather than d(theta)/d(X).

**Reply to comment:** As stated above, we want to just use one way of calculating the ratio between $\Theta$ and $N_2O$. In Fig. 8 *(former Fig. 9)* we use $\Theta$ as the vertical coordinate and the tracer on the x-axis.We added the following text, which states, that we will use only one convention as explained above to just use one way of expressing the ratio to be fully consistent with Fig. 7 and Fig. 8 in the revised manuscript *(former Fig. 8 and Fig. 9)*.

**Changes to manuscript:** Note that we will use in the following the inverse relation $\partial\Theta/\partial N_2O$ to be mathematically be consistent with the profiles shown in Fig. 7 and in Fig. 6 *(former Fig. 8 and Fig. 7)*. We will consequently apply this convention with $\Theta$ in the numerator and $N_2O$ in the enumerator throughout the paper and will keep the same convention for any analysis which follows below.
* * *
**Comment Page 13 Line 7:** Perhaps use the wording "cross-isentropic" somewhere in this description to refer back to the title and previously used terminology. See General Comments above regarding the use of consistent wording.

**Reply to comment:** We changed the sentence.

**Changes in manuscript:** We therefore investigated if tracer gradients with respect to potential temperature $\Theta$ were changed due to the occurrence of gravity wave induced  turbulence leading to cross-isentropic mixing by comparing local tracer profiles upstream and downstream the mountains
* * *
**Comment Page 13 Line 8:** perhaps say "above the tropopause" instead of "at the tropopause" since your measurements are not directly at the tropopause. Figure 8 only shows a diagram of this relationship above the tropopause, so using the same terminology in the text will make it clearer.

**Reply to comment:** We clarified the statement since we want to point out the general properties of $N_2O$ at this point.

**Changes in manuscript:** Since $N_2O$  in the lowermost stratosphere is not affected by local chemistry it is purely under dynamical control
* * *
**Comment Page 13 Line 9:** Your "hypothesis" is difficult to parse from the text due to complex sentence structure - please modify lines 5-7 to more clearly indicate your prediction refers only to the cause of the observed changes to d(X)/d(theta)

(gravity wave induced turbulent mixing). Otherwise your hypothesis could be misidentified as just saying that d(X)/d(theta) changes, which we know already from the data, vs your actual hypothesis of why d(X)/d(theta) changes.

**Reply to comment:** We have changed sentences in lines 5-7 (see above) and also changed the text of our hypothesis.

**Changes in manuscript:** In particular, the gradient change of the conservative tracer $N_2O$ at the tropopause is perfectly suited to test our hypothesis that gravity wave-induced turbulence lead to cross-isentropic mixing. Since $N_2O$ in the lowermost stratosphere at the tropopause is not affected by local chemistry it is purely under dynamical control.
* * *
**Comment Page 13 Lines 9-10:** Change "at the tropopause" to "just above the tropopause" since the data you present in Figure 7 is "just above the tropopause" according to the figure caption.

**Reply to comment:** Here we refer to the gradient change of $N_2O$ at the tropopause as general property of $N_2O$ and changed the text as given above.
* * *
**Comment Figure 8:** Since your discussion in the text refers to d(X)/d(theta), perhaps it would be better to have your diagram in Figure 8 be a diagram of d(X)/d(theta) vs. theta or altitude instead of making the reader infer changes to d(X)/d(theta) from a theta vs N2O plot. You could then compare this diagram with Figure 11 instead of with Figure 9.

**Reply to comment:** We are interested in the scales which are involved to change of the $\Theta$-$N_2O$-relationship. A diagram of e.g. the ratio of two quantities as a function of altitude, $\Theta$ would not bring up the information, which we want to extract - namely the change of the ratio (i.e. the slope) as function of scales.
* * *
**Comment Page 14 Line 1:** Change "This is schematically shown" to "A schematic of our hypothesized changes to d(X)/d(theta) is shown". See General Comment above regarding unspecific use of pronouns at the beginning of sentences.

**Reply to comment:** Changed as suggested.

**Changes in manuscript:**  A schematic of our hypothesized changes of $N_2O$ to $\Theta$ is shown in Fig. 7 *(former Fig. 8)* shows the evolution of the $N_2O$-$\Theta$ profile
* * *
**Comment Page 14 Line 9:** The use of the word "steeper" is confusing in this case - due to the orientation of the axes in Figure 8, the downstream slope looks "steeper" to the eye than the upstream slope because the plot is oriented to show the dependent variable (theta) on the y axis rather than the x axis. To avoid confusion, it would be clearer to say the gradient d(X)/d(theta) is larger upstream. As suggested above, this would be easier to see visually if the diagram in Figure 8 shows d(X)/d(theta) vs theta or altitude rather than theta vs N2O.

**Reply to comment:** Indeed these qualitative expressions should be avoided. We therefore changed the text as below.

**Comment Page 14 Line 10:** Though it follows from the text, it may be good to state explicitly that the vertical gradient decreases due to mixing, rather than just stating that the gradient is higher upstream than downstream.

**Reply to comment:** According to the change related to the preceding comment we clarified the statement.

**Changes in manuscript:** Thus, in case of gravity wave induced turbulent mixing during flight FF09, we expect a

 more rapid decrease of $N_2O$ with increasing $\Theta$ in the inflow region upwind the mountains than at the downstream side of the mountain ridge as an effect of turbulent mixing. The ratio of $\Theta'$ / $N_2O'$ therefore increases from upstream to downstream due to turbulent mixing.
* * *
**Comment Figure 9:** maybe zoom in on the region from 320 K - 340 K to make it easier to see the changing N2O vs theta relationship.

10  **Reply to comment:** We changed the axis scale starting from 308 K so that the data points from the lower flight leg and the chemical tropopause are included as well.

**Changes in manuscript:** Modified figure.

[Figure]
* * *
**Comment Page 15 Line 4:** change "corresponding to the hypothesis described above" to "consistent with our hypothesis that d(X)/d(theta) will be reduced in regions impacted by gravity wave induced mixing". See General Comments above regarding the use of consistent wording.

20  **Reply to comment:** Changed as suggested.

**Changes in manuscript:** As evident from Fig. 8 *(former Fig. 9)* different slopes of $N_2O$ versus $\Theta$ appear on the upstream side, downstream side and above the mountains  consistent with our hypothesis that the vertical gradient (with respect to $\Theta$) $(\partial N_2O/\partial\Theta)$ (and mathematically correct with regard to Fig. 8 (former Fig. 9)
25  $(\partial N_2O/\partial\Theta)^{-1}$) will be changed in regions impacted by gravity wave induced mixing.
* * *
**Comment Page 15 Line 5:** As stated above, please clarify what is meant by a "compact relationship"

**Reply to comment:** We changed the sentence.

**Changes in manuscript:** The $N_2O$-$\Theta$- relation on the upstream side shows a strong decrease of $N_2O$ with potential temperature $\Theta$ and a compact relationship (i.e. a well defined relationship exhibiting only weak scatter).
* * *
**Comment Page 16 Lines 7-8:** remove "as given in detail further below"

**Comment Page 16 Line 9:** after "different scales", add "using the formula"

**Reply to comment:** Changed as suggested.

**Changes in manuscript:**  We analyzed the data for different averaging periods to account for varying perturbation wave lengths and to  analyze the effect at different scales using the formula:
* * *
**Comment Page 16 Line 15:** Perhaps it would be valuable to explain why the slope d(N2O)/d(theta) decreases due to mixing, as up to this point the only "explanation" is that the slope will change, not how it will change or why.

20 **Reply to comment:** The explanation is provided in the discussion of the scheme in former Fig. 8. Mixing between two different air masses will change the gradients of $N_2O$ and $\Theta$ particularly in regions at the tropopause, where the gradients of both quantities change.
* * *
25 **Comment Page 17 Line 4:** Why are wavelengths of 33 km and 4 km selected for Figure 10? Why not show averaging periods corresponding to the spectral peaks in Figure 5 that match the dominant orographic gravity wave frequencies you identified?

**Reply to comment:** In Fig. 9 *(former Fig. 10)* we want to demonstrate how the ratio $\Theta'$/$N_2O'$ change for the three flight segments as a function of averaging time corresponding to a maximum wavelength (i.e. averaging time) to motivate the next
30 figure. The spectral peaks indicate the local dynamics only, while the tracer distribution integrates over the air parcel history.

**Reply to comment:** We added the plots here as suggested. We will however keep the orginal plots in the manuscript, since they better illustrate the method.

[Figure]
* * *
**Comment Figure 11:** Use a clearer label for the y axis than "Slope" (i.e., $\partial\Theta/\partial N_2O$)

**Reply to comment:** We changed the labels and caption to be more consistent throughout the text.

**Changes in manuscript:** Caption of Fig. 10 (former Fig. 11): Scale dependent analysis for different integration times showing the  ratio of $\Theta'/N_2O'$ for different averaging periods (i.e. wavelengths) for upstream (blue), lee (red) and above mountains (black).
* * *
**Comment Figure 11:** Because your analysis is focused on spatial scales, please convert the x-axis label to spatial scales (i.e. km) to facilitate more intuitive comparisons with orographic wave scales identified in the text and in Figure 5. This will also make it easier to understand how these scales correspond to the wavelet coherence plotted in Fig. 12 where scales are converted to km.

**Reply to comment:** We changed the labels and added a spatial scale as upper x-axis as suggested.
* * *
**Comment Figure 11:** As discussed earlier, why not plot d(X)/d(theta) instead of d(theta)/d(X)? This would make it easier to see that the magnitude of d(X)/d(theta) is larger upstream like you discussed on pages 13-14.

**Reply to comment:** We want to use just one expression for the ratio based on the profile (Fig. 8, former Fig. 9) as explained before.
The corresponding diagram with $N_2O'/\Theta'$ is shown here illustrating the transition from the upstream ratio (blue) to the downstream ratio (red) occurring over the mountains (black).

**Changes in manuscript:** see added figure

[Figure]
* * *
**Comment Page 18 Lines 7-9:** It is confusing to identify the slope behavior at "larger wavelengths" and then refer to these dynamics as "at small scales" in the next sentence, as the greatest downstream slope modulation from the upstream slope occurs for the largest averaging times in the figure (i.e., the largest spatial scales). Please use consistent terminology, as referring to the same scale range as both "larger" and "small" from one sentence to the next is needlessly confusing.

**Reply to comment:** We changed the use of the terms according to the more stringent use of the terminology related to ratios , slopes, etc.

**Changes in manuscript:** The downstream impact is  is evident from the different $\Theta'/N_2O'$  ratio at larger wave lengths at the  downwind side compared to the upstream  ratio. The transition between the upstream and downstream ratios occurs at scales <3 km above the mountains. Therefore we conclude that during FF09 mountain waves modified the   $\Theta'/N_2O'$ ratio at small scales  . They induced cross-isentropic turbulent mixing leading finally to changes at large scales downwind the Alps as evident from the ratio $\Theta'/N_2O'$ and finally the vertical gradient $\partial N_2O/\partial\Theta$ (Fig. 6 *(former Fig. 7))*.
* * *
**Comment Figure 12 Caption:** please clarify what variable is plotted by the arrows and what it means for N2O and theta to be phase shifted by 180 degrees

**Reply to comment:** The color code indicates the wavelet coherence as given by equation 2 of the manuscript. We added a link to equation (2). The arrows indicate the relative phasing of the $N_2O$ and $\Theta$ time series; because of the different vertical gradient of $N_2O$ (decreasing values) and $\Theta$ (increasing values) the phase difference between them should be close to 180° (see Fig. 6 *(former Fig. 7))*.

**Changes in manuscript:** We change the caption: Wavelet coherence of $N_2O$ and potential temperature (wavelength = period · flight speed (216 m/s), see eqn. 2).
* * *
**Comment Figure 12 Caption:** Is the 5% significance level the same as the 95% confidence level discussed earlier in the text? If so, please use the same terminology throughout. See General Comments above regarding the use of consistent wording.

**Reply to comment**: Yes, we changed this for consistency.

**Changes in manuscript:** The solid lines show the  95 % confidence level as given in paragraph 2.4.
* * *
**Comment Page 18 Line 12:** Please clarify that this discussion corresponds to Figure 12.

**Reply to comment:** We added it.

**Changes in manuscript:** To identify the leading spatial and temporal scales for the cross-isentropic (i.e. irreversible) mixing of $N_2O$ we analyzed the wavelet coherence between $N_2O$ and $\Theta$ in Fig. 11 *(former Fig. 12)*.
* * *
**Comment Page 18 Line 14:** Scales referenced in the text should be converted to km to be easier to identify in Figure 12 where you have converted the temporal scale sampling to km scales.

**Reply to comment:** Correct. We changed it accordingly.

**Changes in manuscript:** co-vary across different time scales from  1.7-17.3 km (corresponding to 8-80 s).
* * *
**Comment Page 18 Line 14:** Please clarify what the "phase relation" is, how it is plotted in Figure 12, and what it means to have a phase relation that is constant at 180 degrees.

**Reply to comment:** The phase relation is given by the phase between the oscillating $N_2O$ and $\Theta$ as a function of time (e.g. Fig. 6 *(former Fig. 7)*). For opposite vertical altitude gradients of $N_2O$ and $\Theta$ the phase must 180°, if no mixing occurs.

**Changes in manuscript:** Further, the phase relation  between the time series of $N_2O$ and $\Theta$ (see Fig. 6 *(former Fig. 7)*) is almost constant at 180° for scales < 20 km, which one would expect for opposing vertical gradients of $N_2O$ and $\Theta$ in the stratosphere.
* * *
textbfComment Page 19 Line 1: Because the phase relation is not explained, it is unclear what it means or how it relates to previous conclusions in the text. In addition, it is unclear which conclusion you are referring to by saying "the conclusion from the previous upwind slope analysis" - please state this conclusion explicitly and explain how it is confirmed by this analysis.

**Reply to comment:** The sentence of question is a remnant of a previous version. The statement is explained in former p.19, l.7-14. We removed the sentence.

**Changes in manuscript:**
* * *
**Comment Page 19 Lines 3-14:** Perhaps these lines of text can all be part of the same paragraph rather than having 3 paragraphs discussing the same thing in groups of 1-2 sentences.

**Reply to comment:** We connected the two paragraphs.
* * *
**Comment Page 19 Line 3:** please express "time scales < 40 s" in units of km to make them identifiable in Figure 12

**Reply to comment:** We added the km-value. Also in other places.

**Changes in manuscript:** there is low coherence with values lower than 0.7 for time scales  < 8.7 km (< 40 s) accompanied by a breakdown of the phase relation
* * *
**Comment Page 19 Line 5:** Please explain what feature in Figure 12 indicates a "defined phase transition" and how it is distinct from the rest of the plot (phase transitions are not described in terms of phase shift, which is the only explanation given for the meaning of the arrows in Figure 12)

**Reply to comment:** We thank the reviewer for this point. 'transition' is wrong, 'relation ' is correct, since a well-defined phase relation appears downstream at short wavelengths as opposed to the mountain regions (for wavelengths < 10 km.

**Changes in manuscript:** On the downstream side (from 171°E) especially at small periods higher coherence values and defined phase  relations re-establish compared to the above-mountain regime, albeit more variable than at the

upstream side.
* * *
**Comment Page 19 Line 7:** change "matches roughly" to "roughly matches"

**Reply to comment:** We changed the sentence.

**Changes in manuscript:**  consistent with Fig. 10 *(former Fig. 11)* , in upwind regimes with a high coherence
* * *
**Comment Page 19 Line 8:** change "co-vary. The" to "co-vary: the" - you seem to be explaining what it means to co-vary in the next sentence, which is easier to understand if the sentences are combined.

**Comment Page 19 Line 8:** Is the "calculated slope" from Figure 11? If so, please state this as the text here is talking about Figure 12.

**Reply to comment:** Correct. We changed it accordingly.

**Changes in manuscript:** upwind regimes with a high coherence $N_2O$ and the potential temperature $\Theta$ co-vary : the phase relation between them remains constant across scales and the calculated slope (Fig. 11 *(former Fig. 12)*) is unchanged too.
* * *
**Comment Page 19 Line 10:** What is this "new slope relation"? How is it visible in Figure 12? If you are referring back to Figure 11, please say so and quantify this "new slope relation" with a value from the appropriate plot.

**Reply to comment:** The "old slope" is upstream ratio (-0.7 K/ppbv) and the "new ratio" (-2.2 K/ppbv) is the downstream and above mountain ratio from Fig. 10 *(former Fig. 11)* for periods longer than 40 s (> 8.7 km). We also changed "slope relation" to simply "ratio".

**Changes in manuscript:** Downwind a new  ratio reestablishes as a result of mixing above the mountain ridge, but with a defined phase relation again, but different  ratios.
* * *
**Comment Page 19 Lines 12-14:** Please provide more detailed explanations in the text from lines 3-11, as I do not follow how this conclusion is supported by the analysis of Figure 12.

**Reply to comment:** We rephrased the sentence.

**Changes in manuscript:**  We therefore conclude that for short wave length exhibiting low coherence and the breakdown of the phase above the mountains the gravity waves produced turbulence which lead to cross-isentropic mixing. Therefore, the relationship between $N_2O$ and $\Theta$ is the result of gravity wave induced mixing. Since the mixing is cross-isentropic this changed the $N_2O$-$\Theta$ slope, which is evident at the downstream side, where a modified slope establishes (compared to the upwind side).
* * *
**Comment Figure 13 Caption:** change "colors denotes" to "colors denote"

**Reply to comment:** Changed as suggested.

**Changes in manuscript:** Red colors denote a positive flux and blue colors indicate a negative flux.
* * *
**Comment Page 20 Line 1:** is the "cross wavelet transformation" the part of Page 4 line 25 inside { } ? This is the first usage of the term "cross wavelet transformation" as it is not mentioned in section 2.4. See General Comments regarding consistent use of terminology.

**Reply to comment:** The variables inside the brace of equation 1 is called "cross wavelet transformation" and the real part is called "cospectrum of the cross wavelet transformation". We added the term to p.4

**Changes in manuscript:** The wavelet cospectrum $W^{AB}$ of two time series $A$ and $B$ with the wavelet transforms $W^A$ and $W^B$ is defined as the real part of the cross wavelet transformation
* * *
**Comment Page 20 Line 13:** It is not explained why having a temporal resolution of 10 s precludes the analysis of ozone fluxes - please clarify.

**Reply to comment:** The response time of the ozone instrument TE49 which which was used for the campaign is 10 seconds, so we will not resolve fluxes at shorter scales, which are significant according to $N_2O$.
* * *
**Comment Section 4.3:** Figure 14 and its associated discussion would be easier to understand in the context of the spatial scales plotted in Figures 12 and 13 if Figure 14 was discussed in terms of horizontal scales rather than in terms of temporal frequencies. These scales are included in Figure 14 - please modify the discussion here to include the wavelengths in Figure 14 instead of only referring to the frequencies in Hz.

**Reply to comment:** We added time and spatial information to the figures to facilitate comparisons.
* * *
**Comment Page 20 Line 26:** Remove the comma after "both"

**Reply to comment:** Removed as suggested.

**Changes in manuscript:** The slope of the PSD of both $w$ and $\Theta$ turns towards -5/3 for frequencies exceeding  108 m (2 Hz), which can be related to isotropic turbulence.
* * *
**Comment Page 20 Line 27:** change "smaller 0.3 Hz" to "smaller than 0.3 Hz"

**Reply to comment:** Changed as suggested.

**Changes in manuscript:** which show a slope of -5/3 for frequencies smaller  than 721 m (0.3 Hz)
* * *
**Comment Page 20 Line 29:** remove the comma after "range"

**Comment Page 20 Line 29:** Please explicitly state the frequency range you are talking about

**Reply to comment:** Changed as suggested.

**Changes in manuscript:** The transition of geostrophic to isotropic turbulence as indicated by the transition of PSD-slopes occurs in the  wave length range between 271 m to 721 m (corresponding to 0.8 Hz to 0.3 Hz) where the PSD of the vertical wind indicates a source of turbulent energy.
* * *
**Comment Page 22 Lines 1 and 8:** Starting these two paragraphs with "Further support for our hypothesis and our results come from the analysis of . . . " is unclear in both cases - Please state which aspect of your hypothesis is supported by the data in these introductory sentences.

**Reply to comment:** We clarified the sentence.

**Changes in manuscript:** Further support for our hypothesis that mountain wave induced turbulence perturbed the $N_2O$ profile  comes from the analysis of the cubic root of the eddy dissipation rate EDR $= \epsilon^{1/3}$ from the measured 3D-winds
* * *
**Comment Page 22 Line 5:** "v" should also be a subscript in EDR $_{u,v}$

**Comment Page 22 Line 6:** change "when also" to "where"

**Comment Page 22 Line 6:** change "was enhanced" to "was also enhanced"

**Reply to comment:** Changed as suggested.

**Changes in manuscript:** However, the values of  EDR$_{u,v}$ for the horizontal wind components over the mountains are similar to those of the end of the leg,  where EDR$_w$ was also enhanced in the lee of the mountains.
* * *
**Comment Page 22 Line 9:** change "GTG (Graphical Turbulence Guidance)" to "Graphical Turbulence Guidance (GTG)"

**Reply to comment:** We changed it accordingly.

**Changes in manuscript:** Further support for our hypothesis and our results comes from the analysis of the occurrence of mountain wave induced turbulence using the  Graphical Turbulence Guidance (GTG) using ECMWF operational analysis data
* * *
**Comment Page 22 Line 13:** change "upper flight" to "upper flight leg"

**Reply to comment:** We added it.

**Changes in manuscript:** The weak EDR at the upper flight leg in accordance with the weak turbulence occurrence as opposed to the lower leg
* * *
**Comment Page 23 Line 4:** change "activity the" to "activity at the"

**Comment Page 23 Line 4:** change "and propagating" to "that propagates"

**Comment Page 23 Line 5:** change "this observations" to "these observations"

**Reply to comment:** Changed as suggested.

**Changes in manuscript:** The evidence for strong orographic wave activity at the lower level  that propagating to the 10.9 km level serves as the only plausible explanation for  these observations.
* * *
**Comment Page 23 Line 7:** remove "occurrence"

**Reply to comment:** We removed it.

**Changes in manuscript:** The fact that at the higher level the turbulence is weak during the time of flight must be attributed to the time shift between the two flight legs and the high intermittency of turbulence .
* * *
**Comment Page 23 Line 10:** change "gravity wave occurrence" to "gravity waves"

**Reply to comment:** Change as suggested.

**Changes in manuscript:** We present an analysis of high resolution $N_2O$ measurements in the region of orographic gravity  waves over the Southern Alps in New Zealand during the DEEPWAVE 2014 campaign.
* * *
**Comment Page 23 Line 17:** change "Θ also strong" to "Θ, strong"

**Comment Page 23 Line 17:** change "were observed" to "were also observed"

**Reply to comment:** Changed as suggested.

**Changes in manuscript:** Corresponding to the fluctuations of the vertical wind and potential temperature  Θ, strong fluctuations of the tracer $N_2O$ were also observed at the upper flight leg in the region of the occurrence of orographic waves.
* * *
**Comment Page 23 Line 20:** change "gradient above" to "gradient was observed above"

**Reply to comment:** Changed as suggested.

**Changes in manuscript:** Upstream and downstream of the mountain different vertical gradients of $N_2O$ versus potential temperature Θ were observed and enhanced variability of this gradient was observed above the mountains.
* * *
**Comment Page 23 Line 21:** comma after "inert"

**Reply to comment:** We added it.

**Changes in manuscript:** Since $N_2O$ is chemically inert, a change of the $N_2O$-$\Theta$ relation must be due to cross-isentropic mixing effects

5  ____________________

**Comment Page 23 Line 22:** change "ridge showing reversible" to "ridge with reversible"

**Reply to comment:** Changed as suggested.

10  **Changes in manuscript:** A scale dependent slope analysis shows that mixing was initiated over the mountain ridge  with reversible displacements of tracer isopleths and $\Theta$.
* * *
**Comment Page 23 Line 23:** Again, please clarify what is meant by the "compact slope"

15  **Reply to comment:** Please see reply to comment page 15 line 5.
* * *
**Comment Page 23 Line 25:** "The behaviour" - what behavior? Please be specific. (also note the spelling of behavior without
20  a "u" if you prefer to use American English spelling practices)

**Reply to comment:** We modified the sentence.

**Changes in manuscript:**  Mountain wave induced mixing is also consistent with the indication for wave break-
25  ing and momentum deposition above the mountains between the two flight legs.
A scale dependent slope analysis shows that mixing was initiated over the mountain ridge showing reversible displacements
of tracer isopleths and $\Theta$.  These fluctuations must have perturbed the the compact slope of the $N_2O$-$\Theta$ profile via the generation
of turbulence and thus irreversible turbulent cross-isentropic mixing.

30  ____________________

**Comment Page 23 Lines 28-29:** change "occurring potentially previously" to "that may have occurred"

**Reply to comment:** Changed as suggested.

35  **Changes in manuscript:** Still the power spectral energy spectra of $N_2O$ and $\Theta$ with slopes of -5/3 at the smallest scales
can be seen as the result of the turbulence  that may have occurred on this level.
* * *
40  **Comment Page 23 Line 30:** "The tracer conserves the effect" - what tracer, and what effect? Please be specific.

**Reply to comment:** We specified the statement.
**Changes in manuscript:** The tracer distribution conserves the effect of prior occurrence of  highly transient turbulence
occurrence.

45  ____________________

**Comment Page 23 Lines 30-31:** Again, please define what a "compact relation" is.

**Reply to comment:** Please see replies to comment page 15 line 13 and line 14.

**Comment Page 23 Line 30-32:** "At. . . Mahalo et al., 2011)" - divide this sentence into two sentences. You could do this in line 31 by changing "mountains modulating" to "mountains. The modified compact N2O-theta relation also modulates"

**Comment Page 23 Line 32:** change "similar as" to "similar to the mechanism"

**Reply to comment:** We changed it accordingly.

**Changes in manuscript:** At the downstream side a modified compact $N_2O$-$\Theta$ relation establishes as a result of the wave induced turbulence above the  mountains. The reversible air mass displacements induced by the gravity waves similar  to the mechanism described in (Moustaoui et al., 2010; Mahalov et al., 2011).
* * *
**Comment Page24 Line 2:** change "to 0.5" to "to be 0.5"

**Reply to comment:** Changed as suggested.

**Changes in manuscript:** The vertical fluxes of $N_2O$ are estimated to be 0.5 ppbv $\mathrm{m\,s^{-1}}$ corresponding to negative fluxes of $O_3$ of approximately 10 ppbv $\mathrm{m\,s^{-1}}$.
* * *
**Comment Page 24 Line 4:** remove comma after "fact"

**Comment Page 24 Line 5:** remove comma after "shows"

**Comment Page 24 Lines 7-8:** combine this sentence with the previous paragraph

**Comment Page 24 Line 7:** remove comma after "shows"

**Reply to comment:** Changed it accordingly.

**Changes in manuscript:** The fact that the modified relationship prevails downstream of the mountain shows that the turbulence associated with the orographic waves was associated with cross-isentropic mixing. This approach notably differs from local covariance analysis of vertical winds and tracers since it shows that at least part of the kinematic fluxes contributed to a cross-isentropic component
* * *
**Comment Page 24 Lines 9-16:** Use caution introducing new citations in the conclusions - some of these explanations and citations may be better suited to the introduction. The conclusions of your paper should focus specifically on your results.

**Reply to comment:** We added the gravity wave specific references to the introduction. We kept Riese et al., 2012 at the very end, because they motivate a more general aspect of mixing processes relevant for climate projections.
* * *
**Comment Page 24 Line 10:** replace "tropopause region and lower stratosphere" with "UTLS"

**Reply to comment:** Changed as suggested.

**Changes in manuscript:** Diabatic trace gas fluxes are key for understanding the effect of mixing processes on the large scale composition of the  UTLS where they contribute to the mixing induced uncertainty of radiative forcing estimates
* * *
**Comment Page 24 Line 11:** change "and high degree" to "and a high degree"

**Comment Page 24 Line 12:** comma before "regions"

**Comment Page 24 Line 12:** remove "occurrence"

**Reply to comment:** We changed it accordingly.

**Changes in manuscript:** Though the occurrence of orographic waves has strong seasonality and a high degree of transience (Fritts and Alexander, 2003; Rapp et al., 2021), regions of gravity wave activity are hotspots for turbulence  at the tropopause (Alexander and Grimsdell, 2013; Fritts and Alexander, 2003).
* * *
**Comment Comment Page 24 Line 13:** change "this" to "gravity wave induced turbulence"

**Reply to comment:** Changed as suggested.

**Changes in manuscript:** Our data show that  gravity wave induced turbulence can have a persistent effect on the distribution of species and thus a potential forcing impact of radiatively active tracers by changing their isentropic gradients.
* * *
**Comment Comment Page 24 Lines 15-16:** The organization and meaning of the last sentence is unclear.

**Reply to comment:** We reorganized the sentence.

**Changes in manuscript:** By subsequent isentropic transport as part of the stratospheric flow  the impact of radiatively active traces has a strong non-local component downwind  the mountains. Thus, gravity wave induced mixing contributes to the overall mixing induced uncertainty of radiative forcing (Riese et al., 2012).

---

## Author Comment (AC2)

In this study, the authors investigate the effects of gravity wave breaking and the resulting turbulence on mixing around the tropopause. The study is based on a research flight over the Southern Alps during the DEEPWAVE measurement campaign. In-situ measurements of N2O and CO above and below the tropopause, upstream and downstream of the mountains, have been used to diagnose mixing, while gravity waves and turbulence were analysed in detail using temperature and wind measurement data. The authors report breaking gravity waves and air turbulence close to the tropopause, and a resulting alteration in tracer structure, which shows that significant mixing events have occurred in the affected atmospheric regions.

Overall, I believe the results presented in the manuscript are of a very high standard and clearly merit publication in ACP. The effects that gravity waves have on atmospheric composition and dynamics are still poorly understood, but of high relevance for quantifying tracer transport and large scale dynamics of the atmosphere. The main topic of the paper is therefore highly relevant and of considerable interest to the community. The methods and analysis in this work are generally solid and clearly presented, the analysis of cross-isentropic transport is especially detailed. The figures are well prepared. Presentation of results is also very clear in most parts, I would only suggest to make the mathematical notation more consistent in a few places and to clarify the identification of different flight segments discussed in the text (see minor/technical comments).

**Authors response:**
We thank the reviewer for the careful reading and suggestions which helped to improve the paper. We hope that we adequately addressed the key points. The reviewer comments are given in black, our comments are given in blue, text changes in the manuscript in red.

My two more substantial observations are given below, followed by a list of technical corrections and minor suggestions.

**General points and related observations:**
1) The UTLS region is known for sharp tracer gradients, both horizontal and vertical. The horizontal length scales of tracer filaments resulting from various stratosphere-troposphere exchange (STE) processes (like, for example, planetary wave breaking) can be much smaller than the dimensions of the flight pattern considered here. The structure of $N_2O$ distribution in UTLS, as described in this paper, is shaped by STE and can be affected by various STE events. Since the speed of the aircraft is much larger than that of the wind, the air masses sampled downstream of the mountains are most likely not the same air masses as sampled upstream (or are they at least partially the same?), and thus it is possible that these air masses already had different N2O-$\theta$ profiles even before crossing the mountains. It would be good if authors could comment on such a possibility, or argue why it would not significantly alter the tracer gradient analysis results. It was briefly mentioned that aircraft was not flying through tropopause folds, but air with altered tracer structure could have been advected from elsewhere. I realise that this problem may indeed be very hard to address using only data from airborne in-situ measurements, but there are other arguments that could be made. For example, model data showing no complex structures in the typical stratospheric or tropospheric tracers upstream of the mountains before the flight could strengthen the argumentation that leads to the main conclusions of the paper. N2O data would, of course, be best, but ozone and water vapour, which should be available from ECMWF IFS, could already tell a great deal about possible influence of earlier STE events on the observed air masses. Alternatively, dynamical histories of the sampled air parcels and their surroundings could be investigated. There are also a few interesting details in the manuscript that might make this point more relevant:

**Authors response:**
We thank the reviewer for this point, which is central for our analysis. We checked ERA5 reanalysis data for ozone along the flight track in isentropic coordinates at 15:00 UTC (4 hours before our flight). As evident from the ozone cross section one can see, that a) the ozone distribution is almost isentropic, particularly at 330 K and that b) the vertical extent (in isentropic coordinates) of the ozone layers is rather constant upwind the islands. At 18:00 UTC at the time of our measurements ozone at 330 K is almost isentropically distributed. We therefore think, that advection of tracer gradients at 330 K are highly unlikely to have caused the different tracer gradients up- and downwind the mountains at 330 K.

**Changes in manuscript:**
The dynamical structure is mirrored by the ozone distribution from ERA5 data (not shown) along the flight track, which show a rather homogeneous distribution at $\Theta = 330$ K (approximately flight altitude) notably upwind the region of our measurements. a) The mechanism for modification of N2O-$\theta$ relationship by cross-isentropic transport and mixing, as described in Figure 8,

[Figure]

**Figure 1.** Vertical cross section of ozone from ERA5 along the axis of the flight track of interest for 15:00 UTC (left) and 18:00 UTC (right). Note the almost homogeneous ozone distribution at $\Theta = 330$ K.

5  predicts that air above the mountains should include air masses that fall in between the compact upstream/downstream relationships (shaded region in Figure 8b). Therefore, in Figure 9, one would expect to see some black points (observations over the mountains) in between the compact relationships in blue and red (upstream and downstream data). However, the upstream data forms a compact relationship quite distinct from all the remaining higher-altitude flight leg data, especially in the 316-320 ppbv N2O range. Could this be a possible indication that some of the upstream air masses might have a different composition
10  than over-the-mountain/downstream air masses had before being affected by GWs?

**Authors response:**
The reviewer is right with the observation, that at higher isentropes the $N_2O$-$\theta$ relationship differs. If we assume adiabatic flow in the upwind region, which is reasonable for stratospheric conditions within a short time period (hours) the air mass from
15  higher isentropes would not affect the slope change at lower isentropes. Fig. 8 *(former Fig. 9)* shows that the slope between $\theta = 328$ K to $\theta = 334$ K changes from upwind to downwind in the region of orographic wave occurrence. The correlation of both air masses colored by the upwind, downwind above-mountain region consistent with the profile (Fig. 8 *(former Fig. 9)*), as suggested further below (see Fig. reply) highlights, that the composition between the tracers is almost the same upwind and downwind. This is, however not the case for the vertical profiles of CO and $N_2O$ with respect to Theta (see CO profile).
20  Turbulent mixing changes the tracer-Theta relation, with only weak effect (if at all) on the correlation. The fact, that there are no black points scattered between the upwind and downwind distribution might arise a) from the fact, that mixing may be incomplete not having homogenized the mixing ratios from the isentropes involved and b) the varying fractions of high-$N_2O$ from lower isentropes and low-$N_2O$ air from higher isentropes contributes and c) the flight track cannot cover all mixing states and regions with different mixing efficiency. The relative relation between the air masses indicates, that above the mountains
25  the higher $N_2O$ is more prominent at the flight levels compared to the downwind region.

b) Section 4.3 and the conclusions state that certain features of the results "can be seen as the result of the turbulence occurring potentially previously on this level". If the results suggest that the composition of (at least some of) the observed air
30  masses was significantly affected by the previous turbulence/mixing processes, would it not be natural to ask if all the observed air masses were affected equally?

[Figure]

**Figure 2.** Scatter plot of $N_2O$ vs CO for flight FF09 color coded according to the upwind, downwind and above-mountain regions as provided in the manuscript.

[Figure]

**Figure 3.** Profile of CO for flight FF09 (gray) as a function of $\Theta$ color coded according to the upwind, downwind and above-mountain regions as provided in the manuscript.

**Authors response:**
Similar to our comment above we have to accept, that we didn't sample the complete region of mixing. Note further, the transience of the processes: we might have missed the most active time of turbulence occurrence and mixing at the upper flight level. The fact that we observe only weak dynamical indication of turbulence above the mountains, but the slope change compared to the upwind relation is only explainable if we have a mixing processes between the upwind and downwind side. The further downwind, the stronger the effect of the mixing on the tracer distribution. This is illustrated at the lower flight level

(Fig. 3 *(former Fig. 4))* which indicates turbulence occurrence at the flight track above and downwind the mountains, which leads to ongoing mixing also downwind the mountains.

2) After a dynamical process, such as wave breaking, causes cross-isentropic transport and scale breakdown in the tracer structure, tracers are further (mostly isentropically) mixed by molecular diffusion (e.g. Balluch and Haynes, 1997). The N2O-$\theta$ relationship is a great tool for characterising the cross-isentropic transport, but it would also be interesting to see to what extent the air masses that were transported across isentropes are already mixed into surrounding air. Maybe analysing the different air parcel groups from Figure 9 in N2O-CO space could shed some light on that? Or was N2O-CO analysis inconclusive for these air masses?

**Authors response:**
We provided the figure as suggested in the comment above. As can be seen, the distribution confirms the conclusion showing that the part of the flight track above the mountain has contributed to the mixing region (black dots scattered between the main correlation branches).
Two other findings are remarkable:
1) Also the downwind part shows signs of mixing (red data points between the main branches), which is inline with the behavior at the lower flight level shown in Fig. 3 *(former Fig. 4)*, illustrating high variability downwind the mountain.
2) The downwind correlation data appear as part of the upwind correlation data. At first this seems to be puzzling, but noting, that both $N_2O$ and CO can be regarded as passive tracers, the relative relationship between both tracers should not be changed, when mixing occurs (turbulence acts in the same way to CO and $N_2O$). The fact, that cross-isentropic mixing occurred (i.e. changing the tracer-$\Theta$-relation) is untouched by this.

a) Another interesting feature of Figure 9 is that although the downstream air masses occupy roughly the same range of potential temperatures as the upstream ones, they have a much narrower range of N2O concentrations (close to the mean N2O value) with no outlying points in the rest of the upstream N2O range. Could this potentially suggest that the turbulence over the mountains, which the downstream air masses have experienced for just a few hours, has already mixed the affected air masses quite efficiently, and further isentropic mixing (which would normally be slower) is less relevant here? Again, maybe N2O-CO relationship could be used to confirm this?

**Authors response:**
Indeed we think, that the turbulent mixing is much more efficient and later (isentropic) mixing acts on longer scales. Inspecting the $N_2O$-CO relation one could argue that turbulent mixing perturbs the canonical background correlations (as evident by the scatter data point between the main branches in Fig. 5 *(former Fig. 6)* (green) of the original manuscript). The isentropic mixing provides the background branches, which indeed are the result of mixing, but at different regions and on different timescales. However, this is a qualitative argument, but the fact, that those data points, which corresponds to the turbulence occurrence above the mountains and the different tracer-$\Theta$ relations upwind downwind clearly indicate an efficient diabatic (cross-isentropic) mixing process at the flight acting on the background distribution.

**Minor and technical points:**
* * *
**Comment Page 1 Line 21:** The phrase "conserves the effect" is confusing. It is not quite clear to me how an effect itself (as opposed to physical quantities or the results of the effect) can be conserved. This should perhaps be rephrased.

**Reply to comment:** We clarified the sentence. Via turbulent mixing the traces gas distribution is influenced by the turbulence. This influence is present in the trace gases for a certain time after the turbulence occurrence.

**Changes in manuscript:** Despite only weak turbulence during the stratospheric leg, the cross isentropic gradient and the related composition change on isentropic surfaces from upstream to downstream the mountain unambiguously conserves the

5    effect of turbulent mixing by gravity wave activity  on the trace gas distribution prior the measurements.
* * *
**Comment Page 2 Line 1:** "Orographic gravity waves [...] may affect the large scale stratospheric circulation." Clearly, there is still a lot to be learned about orographic GW forcing and the effect they have on the general circulation, but is there really

10    any doubt whether orographic GWs have an effect at all?

**Reply to comment:** True. Changed it accordingly.

**Changes in manuscript:** Orographic gravity waves play an important role for the thermal and dynamical structure of the

15    atmosphere and  affect the large scale stratospheric circulation
* * *
**Comment Page 7 Line 10:** What exactly is meant by "analysed PV"?

20    **Reply to comment:** We clarified it.

**Changes in manuscript:** The tropopause was crossed around 18:40 UTC, as indicated by the sharp decrease of the $N_2O$ mixing ratio and the  PV interpolated along the flight path.

25    _______________________

**Comment Figure 4:** The wave packet seen between 170.1° E and 170.6° E in the higher altitude leg has a very nice and regular vertical wind w and $\theta$ relationship ($\pi$ / 2 phase shift), just as one would expect from linear wave theory. The same longitude range of the lower altitude leg, however, has an interesting $\theta$ structure that does not correspond that well to w. It might be interesting to see if ECMWF IFS predicts similar structures, as these may be related to wave breaking/reflection.

30
**Reply to comment:** The large scale structure similarly exists in ECMWF IFS, but the small scale fluctuations are missing because the model resolution is too coarse. For this flight no wave breaking was predicted by the models.
* * *
35    **Comment Page 9 Line 1:** The word "where" should be replaced with "were".

**Reply to comment:** True. We changed it accordingly.

**Changes in manuscript:** The flight sections of the two southern legs  were strongly affected by orographic waves

40
* * *
**Comment Page 9 Line 2:** Most literature (and the rest of this paper), provide amplitudes as positive numbers, "+/-" should therefore be omitted for consistency. Also, since fluctuations of potential temperature are mentioned, it would be good to provide their amplitude as well.

45
**Reply to comment:** Correct. We changed it accordingly.

**Changes in manuscript:** Both legs show strong fluctuations of the vertical wind component $w$ and the potential temperature $\Theta$ with amplitudes of  2.5 ms$^{-1}$ and 4.5 K, respectively.

**Comment Page 9 Line 7:** The phrase "[...] indicative for at least a kinematic breakdown [...]" should probably be replaced with "[...] indicative of at least a kinematic breakdown [...]". Also, the whole sentence is confusing, it is not quite clear what the word "but" in L8 refers to.

**Reply to comment:** We clarified the sentence and replaced to "but".

**Changes in manuscript:** At the upper level such a breakdown of scales turbulence is not prominent, although the fluctuations of $\Theta$, $w$ and $N_2O$ (Fig. 3 *(former Fig. 4)*) are indicative for of at least a potential kinematic flux of $N_2O$, but with only weakly pronounced small scale variability of $w'$.

**Comment Pages 10-11:** I may have missed something simple or misinterpreted the terms used, but the discussion of observed GWs in Section 3.3 appears to contain contradictory statements. For example, the terms "lower/upper flight leg" seem to refer to lower/higher altitude flight segments in most of the discussion on P10 and P11. Also, the long horizontal wavelength wave mode is stated to be "totally absent in the lower leg" (P10 L4), and "partly seen in VH in Fig. 3 around 17:45" (P10 L3). However, according to Fig. 3, the aircraft was flying at the lower of the two main altitude levels (i.e. flying the "lower leg"?) around 17:45 UTC. Perhaps in some cases "lower/upper leg" refers to lower/higher altitude, and in some cases to downstream/upstream? In any case, I feel that the terms used for flight segment identification should be updated in the entire Section 3.3, so that no guesswork is needed. For example, one might consider only using the term "flight leg" for a straight (geodesic) flight segments, and adopting other terms to refer to longer portions of the flight.

**Reply to comment:** Thank you for your thorough reading. On (P10 L7) there was a confusion which we corrected. We also checked the text to make sure that flight leg always denotes a straight, horizontal part of the flight path. The upper flight leg is at 10.9 km and lower flight leg at 7.9 km. Both flight legs include a downstream, an upstream and an above mountain section.

**Changes in manuscript:** The other, shorter mode in the horizontal wind spectra is well-developed only in the upper lower leg.

**Comment Page 11 Line 3:** Dissipation is indeed a likely explanation for the change in vertical wave energy flux, but one must not forget that GWs are often reflected at the tropopause, which complicates the interpretation of energy fluxes in this region. In any case, the turbulence observations in this paper provide a stronger argument that wave energy is indeed dissipated in the altitude range considered here.

**Reply to comment:** That's right. At least, the analysis of the ECMWF model shows no indication wave reflection during the flight (see also Fig. 1b *(former Fig. 2b)*).

**Comment Caption of Figure 6:** Duplication of the article "the".

**Reply to comment:** True. We removed it accordingly.

**Changes in manuscript:** The lower leg lies entirely in the troposphere as indicated by the the dark gray orange data points of $N_2O$ = 328 ppbv.

**Comment Page 12 Line 14:** Strictly speaking, there is nothing "between $\Theta < 328.1$ K and $\Theta > 326.3$ K", as the two intervals

overlap. I would either write "layer between $\Theta = 328.1$ K and $\Theta = 326.3$ K" or, preferably, "a layer with $326.3$ K $< \Theta < 328.1$ K".

**Reply to comment:** Changed as suggested.

**Comment Page 12 Line 15:** I cannot see any green crosses in Fig. 6, perhaps this should refer to green squares?

**Reply to comment:** Correct. We changed it accordingly.

**Changes in manuscript:**  The intermediate points thus mark a layer between  $326.3$ K $< \Theta < 328.1$ K, where the tracer-tracer diagram indicates mixing between the two branches (green  squares in Fig. 5 *(former Fig. 6))*
* * *
**Comment Page 12 Line 5:** The notation "$\partial$N2O $/ \partial\theta$" as given in L4 is clear, concise and well-defined. Therefore, I cannot see any benefit of subsequently introducing so many different terms (N2O-gradient, N2O-$\theta$ gradient, N2O-$\theta$ slope, decrease of N2O with potential temperature $\theta$, N2O decrease with respect to $\theta$, ...) to refer to essentially the same thing.

**Reply to comment:** We refer to the local idealized $N_2O$-$\Theta$ profile as "$\partial N_2O/\partial\Theta$" (or vertical $N_2O$-gradient with respect to $\Theta$), when speaking about general aspects of the profile, likewise the term $N_2O$-$\Theta$ relationship, when talking about general aspects of their relation (not necessarily the profile). We further changed consistently to "$\Theta'/N_2O'$ ratio" when analyzing the ratio of anomalies relative to the mean and will term this as ratio in general. In some cases we though it is helpful to emphasize certain aspects of this relation related to our hypothesis of tracer changes on isentropes.
* * *
**Comment Page 14 Line 10:** Firstly, notation "$\theta'$ N2O$'$ ratio" is a bit odd. "$\theta$ / N2O$'$ ratio" or "the ratio of $\theta'$ and N2O$'$" would already be better. Secondly, as explained in Section 4.1, the ratio N2O$'$ / $\theta'$ depends on integration time and is therefore mathematically not the same as "N2O-$\theta$ gradient". The fact that the measured gradients do not actually depend on the integration time too much (in a reasonable range of integration times) is a meaningful finding that supports the main claims of the paper. It is hence important not to confuse the readers by implying these two quantities are one and the same.

**Reply to comment:** We thank the reviewer for this comment and kept the "$\Theta'$ / $N_2O'$ ratio" as stated above and removed the term gradient.

**Changes in manuscript:** Thus, in case of gravity wave induced turbulent mixing during flight FF09, we expect a  more rapid decrease of $N_2O$ with increasing $\Theta$ in the inflow region upwind the mountains than at the downstream side of the mountain ridge as an effect of turbulent mixing.
* * *
**Comment Page 18 Line 3:** After inspection of Figure 7, it would seem that both $\theta$ and N2O concentration amplitudes given here are peak-to-peak values, actual amplitude values should be half that.

**Reply to comment:** True. Changed it accordingly.

**Changes in manuscript:** The observed $\Theta$  peak-to-peak variability of 8 K (Fig. 6 *(former Fig. 7))* would correspond to $N_2O = 13$ ppbv, while only 4 ppbv are observed consistent with an impact of diabatic mixing processes changing the upstream relation.
* * *
**Comment Page 18 Line 7:** Same issue as P14 L10. I cannot see a good reason for using primed quantities here and unprimed quantities in L9.

**Reply to comment:** We changed the expressions consistently to the above stated convention.

**Changes in manuscript:** The downstream impact is  is evident from the different $\Theta'/N_2O'$  ratio at larger wave lengths at the  downwind side compared to the upstream  ratio. The transition between the upstream and downstream ratios occurs at scales <3 km above the mountains. Therefore we conclude that during FF09 mountain waves modified the  $\Theta'/N_2O'$ ratio at small scales . . They induced cross-isentropic turbulent mixing leading finally to changes at large scales downwind the Alps as evident from the ratio $\Theta'/N_2O'$ and finally the vertical gradient $\partial N_2O/\partial\Theta$ (Fig. 6 *(former Fig. 7)*).
* * *
**Comment Caption of Figure 12:** Main text discusses various wave time scales, and not length scales. Therefore, the figure should have wave period labels.

**Reply to comment:** To make it comparable to the other figures we changed the values in the text to wavelength.
* * *
**Comment Page 18 Line 14:** Perhaps the authors meant "negative vertical N2O gradient"?

**Reply to comment:** Correct. We rearranged the sentence.

**Changes in manuscript:** Further, the phase relation  between the time series of $N_2O$ and $\Theta$ (see Fig. 6 *(former Fig. 7)*) is almost constant at 180° for scales < 20 km, which one would expect for opposing vertical gradients of $N_2O$ and $\Theta$ in the stratosphere.
* * *
**Comment Page 20 Line 9:** The phrase "periods ranging from 8-16 km" should be replaced with "wavelengths of 8-16 km"

**Reply to comment:** Changed as suggested.

**Changes in manuscript:** The last region shows mainly positive trace-gas fluxes with values up to 0.50 ppbv m s$^{-1}$ at  wavelengths ranging from 8-16 km corresponding to the vertical wind energy maximum around $\lambda_x = 10$ km
* * *
**Comment Page 23 Line 30:** Again, the expression "The tracer conserves the effect of [...]" should perhaps be rephrased.

**Reply to comment:** We rephrased the sentence.

**Changes in manuscript:** The tracer distribution conserves the effect of prior occurrence of  highly transient turbulence occurrence.

---

## Author Comment (AC3)

**Update to Reviews:**

We thank the reviewers for their patience to receive the modified manuscript. We tried hard to follow the suggestions of
both reviewers to be more concise and clear regarding the conventions and terminology for relations between $\Theta$ and $N_2O$.
Therefore we applied some clarifications after the first response to the reviewers had been submitted. These clarifications all
only relate to the terminology, but do not concern methods or data analyses.
We added below these updated final comments with the respective changes to the text as they appear in the revised manuscript
to assure consistency between the revised manuscript and our response. The comments below refer to those four comments of
both reviewers which are affected as indicated below.

**Overall Remarks:**

**Updated authors response:**
We thank the reviewer for the careful reading and the constructive suggestions to improve the accessibility to a broader community. We tried to follow most of the suggestions and hope to have satisfied the criticism.
The major changes are a s follows:
1) We included a definition section to the manuscript to clarify the terminology. We checked the manuscript for a consistent
wording particularly of the multi-word expressions.
We included a definition paragraph to the introduction as given below and adopted the text accordingly
2) We removed Fig. 1 and included the wind information into the former Fig. 2 as suggested.
3) We checked the terminology of the $N_2O$-$\Theta$ relationship and the expressions referring to slopes and ratios. We thereby kept
our original idea to just use one way of expression to quantify slopes and ratios. This is directly deduced from the intuitively
native way of analyzing vertical profiles with potential temperature as y-axis (similar to the discussion of temperature profiles,
which are commonly shown as temperature on the x-axis and $\Theta$ or altitude as y-axis). We therefore wanted to keep the emerging quotient throughout the manuscript with $N_2O$ in the denominator and $\Theta$ in the numerator. We think this facilitates to follow
any discussions instead of introducing inverted relationships. This is consistently done through the paper, independent of the
analysis which relates $N_2O$ to $\Theta$. The reader does not need to link the ratio to a specific analysis step or Figure. We think, this
facilitates the thinking. We added a note on this, when first introducing the scheme in Fig. 8 *(former Fig. 9)*, see new text in
comment to page 13, line 1 below.

**Specific Comments line-by-line:**
* * *
**Comment Page 13 Line 1:** It should be stated much earlier in the text that cross-isentropic fluxes are diabatic. 1 and onward:
the text refers to species gradients as d(X)/d(theta), yet the plotted gradient in Figure 11 appears to be inverted as d(theta)/d(X).
Because the text indicates that the tracer slope changes as a function of theta (instead of saying the theta slope changes as a
function of the tracer), it would be much clearer to plot d(X)/d(theta) rather than d(theta)/d(X).

**Updated changes to manuscript:** The decrease of $N_2O$ in the lowermost stratosphere with respect to $\Theta$ is schematically
shown in Fig. 7 *(former Fig. 8)*. For the following analysis we will use the following conventions: we will express the slope
as ratio of the anomalies $\Theta'/N_2O'$ (according to Eqn. 4) to be consistent with the profile view (as in Fig. 7 and Fig. 8 *(former
Fig. 8 and Fig. 9))*. We will apply this convention with $\Theta$ in the numerator and $N_2O$ in the denominator throughout the following analyses below. We will further use the following terminology:
The term $\Theta$-$N_2O$-relation refers to general aspects of their relation, the term $\Theta'/N_2O'$-ratio (associated with a slope) will be
used when referring to the specific measurements further below. A change of this ratio is directly linked to the change of the

vertical gradient with respect to $\Theta$ ($\partial N_2O/\partial\Theta$).
* * *
**Comment Page 14 Line 9-10 (also reviewer 2):** The use of the word "steeper" is confusing in this case - due to the orientation of the axes in Figure 8, the downstream slope looks "steeper" to the eye than the upstream slope because the plot is oriented to show the dependent variable (theta) on the y axis rather than the x axis. To avoid confusion, it would be clearer to say the gradient d(X)/d(theta) is larger upstream. As suggested above, this would be easier to see visually if the diagram in Figure 8 shows d(X)/d(theta) vs theta or altitude rather than theta vs N2O.

**Updated changes to manuscript:** Thus, in case of gravity wave induced turbulent mixing during flight FF09, we expect a  more rapid decrease of N$_2$O with increasing $\Theta$ in the inflow region upwind the mountains than at the downstream side of the mountain ridge as an effect of turbulent mixing. The vertical N$_2$O profile with respect to $\Theta$ is modified from upstream to downstream due to turbulent mixing.
* * *
**Comment Page 15 Line 4:** change "corresponding to the hypothesis described above" to "consistent with our hypothesis that d(X)/d(theta) will be reduced in regions impacted by gravity wave induced mixing". See General Comments above regarding the use of consistent wording.

**Updated changes to manuscript:** As evident from Fig. 8 *(former Fig. 9)* different  relations between $\Theta$ and N$_2$O appear on the upstream side, downstream side and above the mountains . The different relations are consistent with our hypothesis that the relationship between $\Theta$ and N$_2$O (and consequently the vertical gradient $\partial N_2O/\partial\Theta$) will be changed in regions impacted by gravity wave induced mixing.
* * *
**Comment Page 18 Lines 7-9 (also reviewer 2):** It is confusing to identify the slope behavior at "larger wavelengths" and then refer to these dynamics as "at small scales" in the next sentence, as the greatest downstream slope modulation from the upstream slope occurs for the largest averaging times in the figure (i.e., the largest spatial scales). Please use consistent terminology, as referring to the same scale range as both "larger" and "small" from one sentence to the next is needlessly confusing.

**Updated changes to manuscript:** The downstream impact  is evident from the different $\Theta'/N_2O'$ -ratio at larger wavelengths at the  downwind side compared to the upstream  ratio.  Therefore we conclude that during FF09 mountain waves modified the  $\Theta'/N_2O'$-ratio by the generation of turbulence at small scales . They induced cross-isentropic turbulent mixing leading to changes at large scales downwind the Alps as evident from the $\Theta'/N_2O'$-ratio and finally the vertical gradient $\partial N_2O/\partial\Theta$ (Fig. 6 *(former Fig. 7))*.
* * *
**Comment Page 19 Lines 12-14:** Please provide more detailed explanations in the text from lines 3-11, as I do not follow how this conclusion is supported by the analysis of Figure 12.

**Updated changes to manuscript:**  We therefore conclude that above the mountains the low coherence and the breakdown of the phase relationship at short wavelength were an effect of the gravity waves which produced turbulence and led to cross-isentropic mixing. Therefore, the change in the $\Theta'/N_2O'$-ratio from the upwind to the downwind side is the result of gravity wave induced mixing. Since the mixing is cross-isentropic this changed the $\Theta'/N_2O'$-ratio, which is evident at the downstream side, where a modified ratio establishes (compared to the upwind side).

---

## Editor Decision (ED1)

Editor's comments on revised Gravity wave induced cross-isentropic mixing: A DEEPWAVE case study

p.1 l.18 downwind of the Alps

p.1 l.19 and $N_2O$, are

p.1 l.23 downstream of the mountain

p.1 l. 24 prior to the measurements

p.2 l.5 I think this sentence would read better as 'Orographic gravity waves affect the large-scale stratospheric circulation and play an important role in determining the thermal and dynamical structure of the atmosphere'

p.2 l.10 I don't think the sentence 'Gravity waves … 2010)' is needed – the paragraph reads better without it.

p.2 l. 13 Both types of instability

p.2 l.14 occurs, potentially leading to mixing…

p.2 l.17 radiation-driven

p.2 l.18-25. This part of the text needs redrafting. Do you mean that wave breaking causes diabatic heating (or cooling) separate to its role in promoting turbulence? If so, please explain what process you are referring to. Otherwise the sentence beginning 'Wave breaking' could be omitted and the following sentence changed to 'In addition, turbulence produced by wave breaking and wind shear above the tropopause …….' While I like the term 'cross-isentropic mixing', all mixing driven by turbulence (whether cross-isentropic or not) is irreversible. I'm not sure what point you're trying to make here so won't offer an alternative wording.

p.2 l.23 and 26. Just because a process is diabatic doesn't mean it's irreversible. Radiative heating and cooling for example is in principle reversible, as is condensation and subsequent evaporation of a cloud. You need to be much clearer about the difference between diabatic and irreversible processes.

p.3 l.11 omit 'occurrence'

p.3 l.18 omit 'diabatic'

p.4. l.5 During 12 July, which……paper, no HIAPER…..

p.4 l.9 Herriott cell

p.5 l.17 on 12 July

Fig 1 caption. 'The solid red line in b) denotes the -2 pvu isoline' , also say that the solid black line is the flight track.

p.6.l.7 upwind of the region

p.7 l.1 Particularly in the regions of strong variability of the vertical wind, Ө, $N_2O$……..

p.8 l.1 downwind of

p.13 l.15 downwind of the mountain is indicative of

p.13 l.20 downstream of

p.13 l.22 either 'led' or 'leads' but not 'lead'

p.13 l.28 the following analysis (omit 'below')

p.13 l.30 'referring to specific measurements' (omit 'further below')

Fig 7 Gray arrows mentioned in caption but not visible on diagram

p.16 l.9 omit 'on display'

p.17 l.1 -0.6 K/ppbv  (you have it as +!) (also l.8)

p.19 l.16 Reference is made here to 'the last region' and in the next section to 'the second region'. Which parts of the plot are you referring to in each case?

p.20 l.11. 'frequencies smaller than 721 m' – clearly this is nonsense even if you do put 0.3 Hz in brackets. More importantly, the $E\sim k^{-3}$ spectrum of geostrophic (2-D) turbulence is usually found at much larger scales than here (several hundred km e.g. Nastrom and Gage 1985) so although your results are consistent with $k^{-3}$ for wavelengths 1-10 km it doesn't follow that this is due to 2-D turbulence. I don't agree either that the slopes of both w and $\theta$ 'turn to -5/3' at high frequency – the $\theta$ spectrum in particular could be argued to have the same slope throughout.

p.21 l. 6 you're saying here that the $N_2O$ spectrum has the same slope (-5/3) throughout, which isn't what the diagram shows. You need to be very careful with your argument here. Both $N_2O$ and $\theta$ are passive tracers on the time and length scales of turbulence, so their spectra should look the same. The fact that their slopes are so different suggests either an instrumental issue, probably with $N_2O$, or some sampling issue. I certainly don't see a 'transition from geostrophic to isotropic turbulence' (p21 l.8).

p.21 l/13 cube root, also 'the we used' at the end of the line

p.22 l.1. First of all, it is difficult to distinguish the lines in the EDR panel of fig 14 – could they be coloured? But either way, two of the components are enhanced at the end of the line whereas the third is not. The text says the opposite – that only one component was enhanced. Text and figure must be consistent.

---

## Author Response (AR2)

**Editor's comments on revised Gravity wave induced cross-isentropic mixing: A DEEPWAVE case study**

We thank the editor for the clarifications and suggestions to improve the paper. We hope that we adequately addressed the key points.
* * *
**Comment Page 1 Line 18:** downwind of the Alps

**Comment Page 1 Line 19:** and $N_2O$, are

**Reply to comments:** We changed it as suggested.

**Changes in manuscript:** The $\Theta$-$N_2O$-relation downwind of the Alps modified by the gravity wave activity provides clear evidence that trace gas fluxes, which were deduced from wavelet co-spectra of vertical wind and $N_2O$, are at least in part cross-isentropic.
* * *
**Comment Page 1 Line 23:** downstream of the mountain

**Comment Page 1 Line 24:** prior to the measurements

**Reply to comments:** We changed it as suggested.

**Changes in manuscript:** Despite only weak turbulence during the stratospheric leg, the cross-isentropic gradient and the related composition change on isentropic surfaces from upstream to downstream of the mountain unambiguously conserves the effect of turbulent mixing by gravity wave activity on the trace gas distribution prior to the measurements.
* * *
**Comment Page 2 Line 5:** I think this sentence would read better as 'Orographic gravity waves affect the large-scale stratospheric circulation and play an important role in determining the thermal and dynamical structure of the atmosphere'

**Reply to comment:** We changed as suggested.

**Changes in manuscript:** Orographic gravity waves affect the large-scale stratospheric circulation and play an important role for in determining the thermal and dynamical structure of the atmosphere and affect the large scale stratospheric circulation
* * *
**Comment Page 2 Line 10:** I don't think the sentence 'Gravity waves . . . 2010)' is needed – the paragraph reads better without it.

**Reply to comment:** We removed the sentence.

**Changes in manuscript:** Gravity waves propagate across the UTLS where static stability increases at the tropopause
* * *
**Comment Page 2 Line 13:** Both types of instability

**Comment Page 2 Line 14:** occurs, potentially leading to mixing. . .

**Reply to comments:** Changed as suggested.

**Changes in manuscript:** Both types of  instability may lead to the occurrence of turbulence, particularly when wave breaking occurs , potentially leading to mixing of trace species
* * *
**Comment Page 2 Line 17:** radiation-driven

**Reply to comment:** We added the hyphen.

10

**Changes in manuscript:** To overcome this dynamical barrier diabatic processes are required, which can be associated with radiation-driven processes or phase transitions of water.
* * *
15 **Comment Page 2 Line 18-25. also next comment line 25-26:** This part of the text needs redrafting. Do you mean that wave breaking causes diabatic heating (or cooling) separate to its role in promoting turbulence? If so, please explain what process you are referring to. Otherwise the sentence beginning 'Wave breaking' could be omitted and the following sentence changed to 'In addition, turbulence produced by wave breaking and wind shear above the tropopause . . . . . . .' While I like the term 'cross-isentropic mixing', all mixing driven by turbulence (whether cross-isentropic or not) is irreversible. I'm not sure what
20 point you're trying to make here so won't offer an alternative wording.

**Reply to comment:** As indicated by one of the reviewers we wanted to include a link to the scale breakdown initiated by [planetary] wave breaking and subsequent stirring as opposed to the small scale gravity wave induced turbulence, which is followed in the next sentence.
25 Regarding the terminology we fully agree, that turbulence is irreversible. We want to clarify in this section, that an irreversible exchange across the tropopause can occur via turbulence, and second to emphasize the entropy changing (and thus potential temperature changing) nature of this. The later is directly linked to the observations later, though a redundant statement. However, we think it helps for the later discussions throughout the paper.
Irreversibility is associated with entropy change and thus change of potential temperature, which is also the case for a diabatic
30 process. The opposite however is indeed not true and diabatic is not necessarily irreversible. We sharpened this further.

**Changes in manuscript:** Wave breaking of planetary waves and stirring may cause  a scale breakdown in the tracer structure with subsequent mixing of tracers by molecular diffusion (e.g. Balluch and Haynes, 1997). In addition, turbulence  produced by wave breaking and wind shear above the tropopause (Shapiro,
35 1980; Söder et al., 2021; Kaluza et al., 2021, 2022; Lilly et al., 1974) provides such another efficient diabatic process. It leads to cross-isentropic mixing and thus irreversible trace gas exchange at the tropopause. We will use the term 'cross-isentropic' to emphasize the irreversible (entropy changing i.e. potential temperature changing and therefore diabatic) nature of this process. Further the term 'cross-isentropic' allows to distinguish from 'quasi-isentropic mixing'. The latter is driven by synoptic and planetary waves leading to stirring and mixing best approximated along isentropes.
40  Irreversible (entropy changing) processes lead to a redistribution of tracers which must be therefore cross-isentropic providing a tracer flux crossing isentropes.
* * *
**Comment Page 2 Line 25 and 26:** Just because a process is diabatic doesn't mean it's irreversible. Radiative heating and
45 cooling for example is in principle reversible, as is condensation and subsequent evaporation of a cloud. You need to be much clearer about the difference between diabatic and irreversible processes.

**Reply to comment:** See comment and changes above.
* * *
**Comment Page 3 Line 11:** omit 'occurrence'

**Reply to comment:** We removed it.

**Changes in manuscript:** Direct observations of gravity wave induced cross-isentropic tracer mixing are sparse, since this requires high resolution measurements of passive tracers (i.e. tracers without chemical or microphysical sources or sinks) exactly in the region of turbulence  associated with these waves.

10  _______________________
**Comment Page 3 Line 18:** omit 'diabatic'

**Reply to comment:** We omitted it.

15  **Changes in manuscript:** This study provides evidence on the basis of observed passive tracers that the breaking of mountain waves can lead to  tracer redistribution in the tropopause region by cross-isentropic mixing.
* * *
**Comment Page 4 Line 5:** During 12 July, which......paper, no HIAPER.....

20

**Reply to comment:** We changed it accordingly.

**Changes in manuscript:** During the 12 July, which is the day of the analysis in this paper no HIAPER flight was performed.

25
* * *
**Comment Page 4 Line 9:** Herriott cell

**Reply to comment:** Correct, we changed it.

30

**Changes in manuscript:** The instrument consists of a  Herriott cell with a path length of 36 m.
* * *
**Comment Page 5 Line 17:** on 12 July

35

**Reply to comment:** We changed it accordingly.

**Changes in manuscript:** In this study we focus on research flight FF09 of the Falcon aircraft on the 12 July 2014 starting at 17:15 UTC to 20:15 UTC

40
* * *
**Comment Fig 1 caption:** 'The solid red line in b) denotes the -2 pvu isoline' , also say that the solid black line is the flight track.

**Reply to comment:** True, we changed it.

45

**Changes in manuscript:** The solid red line denotes the -2 pvu isoline and the solid black line denotes the flight track. The black dashed lines denote contours of the horizontal wind velocity (10, 15, 20, 25 m s$^{-1}$ in (a) and 10, 20, 30 m s$^{-1}$ in (b)) and the gray dashed lines in (b) denote contours of potential temperature.
* * *
**Comment Page 6 Line 7:** upwind of the region

**Reply to comment:** Accordingly changed.

**Changes in manuscript:** The flat dynamical tropopause structure is mirrored by the ozone distribution from ERA5 data (not shown), which shows a rather homogeneous distribution at $\Theta = 330$ K (approximately flight altitude) and notably also upwind of the region of our measurements.
* * *
**Comment Page 7 Line 1:** Particularly in the regions of strong variability of the vertical wind, $\Theta$, $N_2O$........

**Reply to comment:** Changed as suggested.

**Changes in manuscript:** Particularly, in the regions of strong variability of the vertical wind also, $\Theta$, $N_2O$ and CO show enhanced variability and strong fluctuations during the stratospheric part of the flight.
* * *
**Comment Page 8 Line 1:** downwind of

**Reply to comment:** We added it.

**Changes in manuscript:** However, its variability does increase downwind of the mountains similar to $w$ and $\Theta$, indicating the occurrence of turbulence.
* * *
**Comment Page 13 Line 15:** downwind of the mountain is indicative of

**Reply to comment:** We changed it as suggested.

**Changes in manuscript:** Therefore, a change of tracer gradient (i.e. $\partial N_2O/\partial\Theta$) as a function of $\Theta$ downwind of the mountain is indicative for of irreversible cross-isentropic tracer exchange which might have occurred above the mountain ridge.
* * *
**Comment Page 13 Line 20:** downstream of

**Reply to comment:** We added it.

**Changes in manuscript:** We therefore investigated if tracer gradients with respect to potential temperature $\Theta$ were changed due to the occurrence of gravity wave induced turbulence leading to cross-isentropic mixing by comparing local tracer profiles upstream and downstream of the mountains ($\partial\chi/\partial\Theta|_{up} \neq \partial\chi/\partial\Theta|_{down}$).
* * *
**Comment Page 13 Line 22:** either 'led' or 'leads' but not 'lead'

**Reply to comment:** Correct. We changed it.

**Changes in manuscript:** In particular, the gradient change of the conservative tracer $N_2O$ at the tropopause is perfectly suited to test our hypothesis that gravity wave-induced turbulence lead led to cross-isentropic mixing.
* * *
**Comment Page 13 Line 28:** the following analysis (omit 'below')

**Reply to comment:** We omitted it.

**Changes in manuscript:** We will apply this convention with $\Theta$ in the numerator and $N_2O$ in the denominator throughout the following analyses .
* * *
**Comment Page 13 Line 30:** 'referring to specific measurements' (omit 'further below')

**Reply to comment:** We removed it.

**Changes in manuscript:** The term $\Theta$-$N_2O$-relation refers to general aspects of their relation, the term $\Theta'/N_2O'$-ratio (associated with a slope) will be used when referring to the specific measurements .
* * *
**Comment Fig 7:** Gray arrows mentioned in caption but not visible on diagram

**Reply to comment:** In the last update we removed the gray arrows. We changed the caption.

**Changes in manuscript:** Schematic evolution of potential temperature $\Theta$ versus $N_2O$ in the presence of cross-isentropic mixing  at the tropopause (e.g. by orographic wave-induced turbulence).
* * *
**Comment Page 16 Line 9:** omit 'on display'

**Reply to comment:** We removed it.

**Changes in manuscript:** The Figure shows the same subset of data using averaging periods  of 150 s (Fig. 9(a)) and 20 s (Fig. 9(b)) corresponding to a horizontal scale of 33 km and 4 km, respectively.
* * *
**Comment Page 17 Line 1:** -0.6 K/ppbv (you have it as +!) (also l.8)

**Reply to comment:** True. We changed it in both places.

**Changes in manuscript:** With a value of about -0.6 K/ppbv the ratio is almost constant over all averaging periods.
* * *
**Comment Page 19 Line 16:** Reference is made here to 'the last region' and in the next section to 'the second region'. Which parts of the plot are you referring to in each case?

**Reply to comment:** Correct. We changed it accordingly.

**Changes in manuscript:** The first region is between 170.0°E and 170.6°E, the second  between 170.8°E and 171.0°E. The  first region shows mainly positive trace-gas fluxes with values up to 0.50 ppbv m s$^{-1}$ at wavelengths ranging from 8-16 km corresponding to the vertical wind energy maximum around $\lambda_x = 10$ km

**Comment Page 20 Line 11:** 'frequencies smaller than 721 m' – clearly this is nonsense even if you do put 0.3 Hz in brackets. More importantly, the E~$k^{-3}$ spectrum of geostrophic (2-D) turbulence is usually found at much larger scales than here (several hundred km e.g. Nastrom and Gage 1985) so although your results are consistent with $k^{-3}$ for wavelengths 1-10 km it doesn't follow that this is due to 2-D turbulence. I don't agree either that the slopes of both w and Θ 'turn to -5/3' at high frequency – the Θ spectrum in particular could be argued to have the same slope throughout.

**Reply to comment:** We apologize for the incorrect assignments. We also agree, that a slope of -3 is indicative for geostrophic turbulence at large scales. We therefore changed the text accordingly. Regarding the slopes also see reply and changes to the next comment.

**Changes in manuscript:** The power spectral density (PSD) for  Θ and the vertical wind component $w$ in Fig. 13 show a slope of -3 for  wavelengths longer than 721 m (< 0.3 Hz),  in agreement with the observations during airborne measurements during START08 (Zhang et al., 2015) for wavelengths < 10 km. For  shorter wavelengths (i.e. higher frequencies) the increase of the power spectral density at 433 m (0.5 Hz) would be consistent with a potential source of turbulent energy above the mountains.
* * *
**Comment Page 21 Line 6:** you're saying here that the $N_2O$ spectrum has the same slope (-5/3) throughout, which isn't what the diagram shows. You need to be very careful with your argument here. Both $N_2O$ and Θ are passive tracers on the time and length scales of turbulence, so their spectra should look the same. The fact that their slopes are so different suggests either an instrumental issue, probably with N2O, or some sampling issue. I certainly don't see a 'transition from geostrophic to isotropic turbulence' (p21 l.8).

**Reply to comment:** We thank for this comment, which we took for a piecewise analysis of the slopes: Focusing on Θ and $N_2O$ we derive for the frequency interval 0.015 Hz - 0.3 Hz -2.8 ($N_2O$) and -3.2 (Θ). For the high frequencies (> 0.9 Hz) we get -1.3 ($N_2O$) and -1.8 (Θ). For vertical wind we used only the frequencies > 1.9 Hz due to the local maximum around 0.6 Hz and derive a slope of -1.9. For clarity Fig. 1 shows the PSD for $N_2O$, Θ and vertical wind separately.
We have no indications for technical problems with the $N_2O$ measurement system nor clear indication, that we are approaching the white noise limitation of the measurement at highest frequencies. Different slopes between the passive tracers $N_2O$, $O_3$ and the potential temperature are described in Bacmeister (1996). A similar behavior (with a flattening slope at high frequencies) was reported by Zhang et al. (2015) for pressure. It is interesting to note that in their measurements the spectral behavior did not follow potential temperature and that the flattening slope of static pressure was reported for a flight segment, where the vertical wind showed a source of turbulence (their section M2), which is similar to our observation. Nonetheless, since we can't fully exclude an artifact, we changed the relevant text passage and also removed the mentioning of a transition of geostrophic to isotropic turbulence.

**Changes in manuscript:**  For wavelengths below 108 m (> 2 Hz), the slope of the PSD of both $w$ and Θ approach -5/3, which can be related to isotropic turbulence. Fig. 13 also shows the PSD of $N_2O$ . It has a slope of -3 for longer wavelengths (smaller frequencies) and a flattening slope for wavelengths smaller than 721 m (> 0.3 Hz).  A change of the turbulent behavior as indicated by the transition of PSD-slopes occurs in the wavelength range between 271 m to 721 m (corresponding to 0.8 Hz to 0.3 Hz), where the PSD of the vertical wind indicates a source of turbulent energy. Notably the PSD of $N_2O$ indicates towards a turbulent behavior for  small wavelengths and thus the occurrence of turbulent fluxes corresponding to the analysis in the previous section.

[Figure]

**Figure 1.** Power spectral density for $N_2O$ (left; black), potential temperature (middle; orange) and vertical wind (right; blue) for the flight segment above the mountains. The power spectral density has been smoothed by a boxcar average of 5 seconds. The green and red reference lines have slopes of -5/3 and -3.
* * *
**Comment Page 21 Line 13:** cube root, also 'the we used' at the end of the line

**Reply to comment:** We changed it accordingly.

**Changes in manuscript:** Further support for our hypothesis that mountain wave induced turbulence perturbed the $N_2O$ profile comes from the analysis of the  cube root of the eddy dissipation rate EDR = $\epsilon^{1/3}$ from the measured 3D-winds. For this analysis  we used the method by

10
* * *
**Comment Page 22 Line 1:** First of all, it is difficult to distinguish the lines in the EDR panel of fig 14 – could they be coloured? But either way, two of the components are enhanced at the end of the line whereas the third is not. The text says the opposite – that only one component was enhanced. Text and figure must be consistent.

15 **Reply to comment:** We changed the colors of the figure (Fig. 2) and clarified the text.

**Changes in manuscript:** However, the values of $EDR_{u,v}$ for the horizontal wind components are enhanced over the mountains . In the lee of the mountains the EDR of all wind components is enhanced.

20

[Figure]

**Figure 2.** Time series of (a) vertical wind, (b) EDR for the measured wind components for the upper flight leg indicating weak, but non-vanishing turbulence during the time of flight above the Southern Alps (orography shown in (c)).

**References**

Bacmeister, J. T.: Stratospheric horizontal wavenumber spectra of winds, potential temperature, and atmospheric tracers observed by high-altitude aircraft, Journal of Geophysical Research Atmospheres, 101, 9441–9470, https://doi.org/10.1029/95JD03835, 1996.

Zhang, F., Wei, J., Zhang, M., Bowman, K. P., Pan, L. L., Atlas, E., and Wofsy, S. C.: Aircraft measurements of gravity waves in the
5    upper troposphere and lower stratosphere during the START08 field experiment, Atmospheric Chemistry and Physics, 15, 7667–7684, https://doi.org/10.5194/acp-15-7667-2015, 2015.